# Non-Small-Cell Lung Cancer Signaling Pathways, Metabolism, and PD-1/PD-L1 Antibodies

**DOI:** 10.3390/cancers12061475

**Published:** 2020-06-05

**Authors:** Mariacarmela Santarpia, Andrés Aguilar, Imane Chaib, Andrés Felipe Cardona, Sara Fancelli, Fernando Laguia, Jillian Wilhelmina Paulina Bracht, Peng Cao, Miguel Angel Molina-Vila, Niki Karachaliou, Rafael Rosell

**Affiliations:** 1Department of Human Pathology “G. Barresi”, Medical Oncology Unit, University of Messina, 98122 Messina, Italy; c.santarpia@hotmail.com; 2Instituto Oncológico Dr Rosell, Hospital Universitario Quirón-Dexeus, 08028 Barcelona, Spain; aaguilar@oncorosell.com; 3Institut d’Investigació en Ciències de la Salut Germans Trias i Pujol (IGTP), 08916 Badalona, Spain; imanchaib@gmail.com (I.C.); sfancelli.sf@gmail.com (S.F.); flaguia@igtp.cat (F.L.); 4Foundation for Clinical and Applied Cancer Research-FICMAC Translational Oncology, Bogotá 100110, Colombia; a_cardonaz@yahoo.com; 5Pangaea Oncology, Hospital Universitario Quirón-Dexeus, 08028 Barcelona, Spain; Jill94bracht@gmail.com (J.W.P.B.); mamolina@panoncology.com (M.A.M.-V.); 6College of Pharmacy, Nanjing University of Chinese Medicine, Nanjing 210023, China; cao_peng@njucm.edu.cn; 7Merck KGaA, 64293 Darmstadt, Germany; niki.karachaliou@merckgroup.com

**Keywords:** anti-PD-1/PD-L1 monoclonal antibodies, inflammation-associated cell death pathways, K-Ras mutations, LKB1 mutations, metabolic rewiring

## Abstract

Treatment of advanced (metastatic) non-small-cell lung cancer (NSCLC) is currently mainly based on immunotherapy with antibodies against PD-1 or PD-L1, alone, or in combination with chemotherapy. In locally advanced NSCLC and in early resected stages, immunotherapy is also employed. Tumor PD-L1 expression by immunohistochemistry is considered the standard practice. Response rate is low, with median progression free survival very short in the vast majority of studies reported. Herein, numerous biological facets of NSCLC are described involving driver genetic lesions, mutations ad fusions, PD-L1 glycosylation, ferroptosis and metabolic rewiring in NSCLC and lung adenocarcinoma (LUAD). Novel concepts, such as immune-transmitters and the effect of neurotransmitters in immune evasion and tumor growth, the nascent relevance of necroptosis and pyroptosis, possible new biomarkers, such as gasdermin D and gasdermin E, the conundrum of K-Ras mutations in LUADs, with the growing recognition of liver kinase B1 (LKB1) and metabolic pathways, including others, are also commented. The review serves to charter diverse treatment solutions, depending on the main altered signaling pathways, in order to have effectual immunotherapy. Tumor PDCD1 gene (encoding PD-1) has been recently described, in equilibrium with tumor PD-L1 (encoded by PDCD1LG1). Such description explains tumor hyper-progression, which has been reported in several studies, and poises the fundamental criterion that IHC PD-L1 expression as a biomarker should be revisited.

## 1. Introduction

The programmed death-1 (PD-1) pathway is a key mediator of local immunosuppression in the tumor microenvironment (TME), also modulating T cell priming against tumor antigens and secondary lymph nodes [1]. Blocking PD-1 pathway by inhibiting the PD-1 receptor on immune cells or the PD-L1 ligand on tumor and/or immune cells can inhibit tumor growth and potentially lead to curability. The first study was carried out in 39 patients with metastatic melanoma, colorectal cancer, castrate-resistant prostate cancer, non-small-cell lung cancer (NSCLC) or renal cell cancer who received a single intravenous infusion of anti-PD-1 (MDX-1106, hereafter named nivolumab). One durable complete response, and two partial responses were seen, and two additional patients (with melanoma and NSCLC) showed significant tumor regressions. The serum half-life of anti-PD-1 was 12 to 20 days. Pharmacodynamic assessment indicated a sustained mean occupancy of > 70% of PD-1 molecules on circulating T cells > 2 months following infusion. In nine patients examined, tumor cell surface B7-H1 (PD-L1) expression seems to correlate with the likelihood of response [2]. Currently three monoclonal antibodies that block PD-1 (nivolumab, pembrolizumab and cemiplimab) and three that block PD-L1 (atezolizumab, durvalumab and avelumab) are approved for use by the US Food and Drug Administration (FDA) as first and/or later line treatment for 17 different types of advanced cancers (Table 1) [1]. Cemiplimab (3 mg per kilogram of body weight) every 2 weeks has been used for metastatic cutaneous squamous cell carcinoma with 47% of response. Adverse events include diarrhea, fatigue, nausea, constipation and rash [3]. Cemiplimab alone or in combination with radiotherapy and/or low dose cyclophosphamide have shown a similar safety profile, also the most common treatment-emergent adverse events (TEAEs) were fatigue (45%), nausea (36.7%) and vomiting (25%). The most common immune adverse related events (irAEs) were, arthralgia (10%), hypothyroidism (8.3%) and maculopapular rash (8%). The side effects are comparable with other anti-PD-1 agents. Two complete responses and seven partial responses were observed among 60 patients [4].

NSCLC still has a poor prognosis and immunotherapy (IMT) has become part of the treatment for patients without driver alterations (epidermal growth factor receptor, EGFR, or anaplastic lymphoma kinase, ALK). The ASCO and OH Joint Panel Guideline recommend pembrolizumab for non-squamous cell carcinoma (non-SCC) with high PD-L1 expression (tumor proportion score [TPS] ≥ 50%) [5]. Other recommended options are, the combination of pembrolizumab with carboplatin and pemetrexed, or atezolizumab with carboplatin, paclitaxel and bevacizumab or atezolizumab, carboplatin and nab-paclitaxel. For most patients with non-SCC and negative or low positive PD-L1, the recommendation is pembrolizumab, carboplatin and pemetrexed. However, other combinations are also acceptable. Again, for patients with high PD-L1 expression (TPS ≥ 50%) and SCC, the panel recommends single agent pembrolizumab. However, for patients with SCC and negative or low positive PD-L1, it is also permissible, pembrolizumab, carboplatin and paclitaxel or nab-paclitaxel. The recommendations are based on studies with clear information (Impower 130, KEYNOTE 189, KEYNOTE 142, Impower 150 and KEYNOTE 407) [5]. The review of these studies indicates that the response rate is less than 30%, the median progression-free survival (PFS), not exceeding 6 to 8 months, and median overall survival (OS) of 20 months. Therefore, there is an urgent need to understand the mechanisms of immune evasion and how PD-1 or PD-L1 monoclonal antibodies can be combined with other reagents that can circumvent resistance.

There has been a long debate about the predicted role of PD-L1. PD-L1 immunohistochemistry (IHC) assays estimate the percentage of tumor cells with an intensity of membranous expression (TPS and the percentage of immune cells with similar expression). Currently, four PD-L1 assays are FDA approved in lung cancer. The predictive value of these assays is limited, as benefit is also seen in patients whose tumors do not express PD-L1 and, often, no benefit is observed in patients with PD-L1 expression [6]. Molecular genotyping of NSCLC is becoming more frequently implemented. For instance, KRAS mutations co-occur with other alterations and mutations, particularly in serine/threonine kinase 11 (STK11), also known as liver kinase B1 (LKB1), and Kelch-like ECH-associated protein 1 (KEAP1) (see below). TP53 mutations can co-occur, at different frequencies, with other driver alterations, including EGFR mutations, KRAS mutations, MET exon 14 skipping mutations, but also ALK and ROS1 rearrangements [7]. Multiple endeavors have been performed to find the best way to predict response to anti-PD-1 and anti-PD-L1 monoclonal antibodies, including the correlation with tumor mutation burden (TMB) by whole exome sequencing, considering activating mutations in receptor tyrosine kinase mutations, smoking-related mutational signature and human leukocyte antigen status in order to more accurately predict response [8]. In this complex assessment, differences in PD-L1 expression between responders and non-responders were not identified. The presence of RTK mutations (EGFR mutations) was a negative predictor of response [8].

## 2. Anti-PD-1 and Anti-PD-L1 Antibodies and Driver Alterations in NSCLC

Many attempts have been made with anti-PD-1 and anti-PD-L1 antibodies to improve the response and PFS in NSCLC patients with driver alterations. A study was carried out in 551 patients treated in 24 centers from 10 countries. Most patients received nivolumab (466) pembrolizumab (48), atezolizumab (19), durvalumab (11) and the rest, other drugs. Most patients received anti-PD-1 anti-PD-L1 antibodies as second or third-line therapies. The molecular alterations included KRAS (2071), EGFR (125), BRAF (43), MET (36), HER2 (29), ALK (23), RET (16) and ROS1 (7). The median PFS was very short for each of these categories, ranging from 2.1 months to 3.2 months. The reasons for the lack of activity are not well-known [9]. Intriguingly, it has been seen that driver fusions in lung adenocarcinomas co-occur with SETD2 mutations (16% of cases) in contrast with lung adenocarcinomas with EGFR, KRAS, BRAF and MET mutations, where the frequency of SETD2 mutations is only 2% [10]. SETD2 is a histone and microtubule methyltransferase and is considered a tumor suppressor gene playing a critical function in DNA damage repair and remodeling of mitotic spindles. SETD2 mutations partly explain the resistance to PD-1 and PD-L1 antibodies in lung adenocarcinomas driven by fusions, making the screening for SETD2 mutations advisable. SETD2 directly methylates signal- transducer and activator of transcription 1 (STAT1) on K525, that is warranted for the activation of STAT1 and the interferon signaling pathway [11]. These findings warrant further research in the subclasses of NSCLC driven by ALK, ROS, RET and other fusions. Moreover, the interferon signaling hyperactivation can also result in resistance to anti-PD-1 and anti-PD-L1 antibodies, as well as involving auto-immune disease, such as, systemic lupus erythematosus [12]. Reduction in circular RNAs in peripheral blood mononuclear cells (PBMCs) is observed in patients with systemic lupus erythematosus that is accompanied by increased RNase L activity and enhanced protein kinase R (PKR) activation and expression of interferon (IFN)-induced genes. These findings could be relevant for further assessment of the participation of circular RNAs in immune response, knowing that circular RNAs retain PKR. The release of PKR could have a role in the control of viral infections. In normal cells, circular RNAs sequester PKR while viral infections activate RNase L and RNase L cleavage of circular RNA releases PKR [13,14]. Of interest is the fact that stimulator of interferon genes (STING) activates IFN and double stranded RNAs (dsRNAs), with increased sensor levels of MDA5, RIG-1 and PKR [15].

New hints break the paradigm that EGFR tyrosine kinase inhibitors (TKIs) negate the effects of anti-PD-1 and anti-PD-L1 monoclonal antibodies. Recent observations have shown that HypoTKI (hypofractionated EGFR TKI: high doses with low frequency treatment) appears to be more effective than the standard treatment of HyperTKI (hyperfractionated EGFR TKI: low doses with daily treatment). Mice bearing TUBO tumors treated with afatinib with HypoTKI regimen were more effective in reducing tumor burden than HyperTKI. Similar to afatinib, HypoTKI, gefitinib and osimertinib, were more potent than, Hyper TKI, gefitinib or osimertinib, in reducing tumor burden and limiting tumor relapse. It was noted that HypoTKI, but not HyperTKI, increases CD3+, CD8+ and CD4 + T and B cells in the TME. Hypo EGFR TKI can induce hypoptosis in tumor cells, releasing tumor-derived danger-associated molecular patterns (DAMPs), that can activate cGAS-STING and Toll-like receptors (TLR)-Myd88, that are essential for type I interferon production. It is inferred from these findings that hypo-fractionated regimens can be applied to other driver alterations, such as, ALK, opening a new opportunity to re-visit the therapeutic approach of combination of EGFR TKIs with anti-PD-1 and anti-PD-L1 antibodies [16].

## 3. Anti-PD-1 and Anti-PD-L1 Antibodies and Endocytosis

Another strategy to enhance the efficacy of anti-PD-1 and/or anti-PD-L1 monoclonal antibodies could be the inhibition of endocytosis. Endocytosis is crucial in regulating cell-surface expression of a large number of membrane molecules, including signaling receptors involved in anti-tumor immune responses. Clathrin-mediated endocytosis (CME) and clathrin-independent endocytosis (CIE) are responsible for receptor internalization [17]. In the CME, the binding of adaptor protein 2 (AP2) complex to the activated receptor on the cell membrane surface, permits the recruitment of clathrin, which creates a coat around the vesicle in formation. Dynamin GTPase is involved in detaching vesicles from the membrane, and the receptors finally internalized, can be recycled through recycling endosomes and come back to the membrane external surface, or be destroyed by lysosomes. The CIE does not involve clathrin, however, dynamin may be involved [18]. These mechanisms make most receptors, such as, as EGFR and PD-L1, unavailable to be targetable by monoclonal antibody (mAb) and, in cancer cells, they represent escaping mechanisms [19]. The external portion of mAb, called fragment crystallizable region (Fc), interacts with effector cells, such as, NK cells, neutrophils, macrophages, monocytes, eosinophils, and dendritic cells (DCs) to activate the antibody-dependent cellular cytotoxicity (ADCC). The main mechanism of cell death utilized in ADCC is though the release of granzymes and perforine, instead of Fas signaling and the release of reactive oxygen species (ROS) [20].

Based on these assumptions, endocytosis inhibitors can be used to move tumor cell antigens targeted by therapeutic monoclonal antibodies to the cell surface, in order to improve the ADCC and clinical responses to these agents. An ex vivo human tumor assay has shown distinct patterns of EGFR trafficking in SCC, correlating with therapeutic outcomes [21]. The study shows that tumors can be classified into those where EGF was, or was not, able to be endocytosed. Patients in whom tumor EGFR escapes endocytosis respond better to EGFR monoclonal antibody therapy [22]. For decades, the blockage mechanisms of the different patterns involved in endocytosis were investigated without successful results [23]. In this review the author explored the activity of many inhibitors, including those of the CME, such as, potassium depletion, hypertonic sucrose, cytosolic acidification, monodansylcadaverine andphenylarsine oxide (PAO). However, due to side effects of each of these compounds, they are not applicable in vivo.

Dynamin is involved in EGFR- and PD-L1 endocytosis. A reversible small molecular weight inhibitor of dynamin, dyngo4A, shows its efficacy if added to anti-EGFR mAb. The drug combination increases EGFR expression on the cell surface and, at the same time, ADCC is both responsive and refractory in EGFR SCC lines. Dyngo4a also increases the expression of p-Akt, which is involved in mTOR phosphorylation of PRAS40, a powerful inhibitor of mTORC1 complex. This inhibition translates into a blockage of cell transcription [24]. Despite promising results in preclinical studies, the agent has not yet been tested in clinical trials.

In addition to dyngo4a, prochlorperazine (an antiemetic and anti-psychotic drug) is also a dynamin inhibitor that concentrates in cell membranes and can bind to multiple cellular targets [25]. In addition, prochlorperazine increases the interaction between NK cells and cancer cells with a “zippering” effect. Besides blocking CME, prochlorperazine also has a blocking effect in another way, by internalizing receptors, such as, EGFR, HGFR, VEGFR and PDGFR, called fast endophiline-mediated endocytosis (FEME) [24], a fast-acting tubulovesicular endocytic pathway independent from AP2 and clathrin [26].

Prochlorperazine (PCZ) alone, and in combination with anti-EGFR in sensitive and resistant EGFR cells lines, decreases the expression of p-ERK and p-Akt. The lack of phosphorylation of Akt reduces the activity of Bcl-2/Bcl-X complex and NF-κB via apoptosis and gene transcription, respectively. In NSG mice models with the addition of an HLA-II mediated immunity, the combination of PCZ and Cetuximab in EGFR resistant tumors has a statistically significant tumor regression with durable response. This model was reproduced adding avelumab (an anti-PD-1 monoclonal antibody with low ADCC activity) to PCZ in mouse colon carcinoma, showing a significant improvement in inhibition of primary tumor growth compared to PCZ-or avelumab-only treatment. Unfortunately, due to a loss of PD-L1 expression and a reduction of MCH-I molecule expression, the same activity was not shown when the combination is applied to renal cancer cells in the same mice model. A pilot clinical study with five patients shows an increased expression of EGFR on cell surface after 30 min of PCZ infusion without any significant changes in vital signs [24].

Prochlorperazine could be repurposed to enhance the efficacy of anti-tumor mAbs. It is tempting to speculate that FAK inhibitors, or drugs inhibiting FAK, like dihydroartemisinin (DHA) [27], could also be used as dynamin inhibitors (Table 2). Of note, a phase 1/2 study (NCT02758587) is ongoing to assess safety, tolerability and preliminary activity of defactinib (a FAK inhibitor) combined with pembrolizumab in patients with advanced solid tumors, including NSCLC, pancreatic cancer, and mesothelioma. In addition, other markers involved in Src pathway endocytosis control could play an important role, such as the ubiquitin ligase Hakai [22].

## 4. Inhibitor of DNA Binding I (ID1), a Read-Out of Loss of LKB1 and a Predictor of Sensitivity to Gefitinib

ID1 overexpression sensitizes EGFR or KRAS mutant NSCLC cells to gefitinib (EGFR TKI). ID1 overexpression significantly increases the cytotoxicity of gefitinib, independently of the EGFR mutational status of NSCLC [28]. ID1 overexpression has minimal effect in inducing inflammasome execution, such as caspase-1 cleavage, and interleukin-1β (IL-1β) and in induction of gasdermin D (GSDMD), the pyroptosis executor. However, ID1 overexpression activates the pathway related to necroptosis. Phosphorylation of receptor-interacting protein kinase-3 (RIPK3) and phosphorylation of mixed-lineage kinase domain-like protein (MLKL) in gefitinib treated NSCLC cells overexpressing ID1, results in the disruption of the plasma membrane [28]. ID1 overexpression in NSCLC increases mRNA and protein expression of RIPK3 and MLKL. Intriguingly, the tumor suppressor LKB1 is mutationally inactivated in KRAS mutant NSCLC. LKB1 phosphorylates and activates members of the AMPK family, including the salt-inducible kinases (SIKs) that modulate gene expression via inhibitory phosphorylation of cAMP-regulated transcriptional coactivators (CRTCs). The loss of LKB1 causes SIK inactivation and the induction of CRTCs, leading to the upregulation of cAMP response element-binding protein (CREB). It was identified that CRTC2 is upregulated in LKB1-deficient NSCLC, promoting tumor growth via the induction of ID1 [29]. ID1 is a bona fide CREB target gene and its overexpression confers poor prognosis in LKB1 deficient NSCLC. It was suggested that inhibitors of CRTC2 and ID1 could be of great benefit [29]. It is tempting to speculate that gefitinib could be an ID1 inhibitor in LKB1 mutated KRAS NSCLC tumors (Table 2). Patients with LKB1 mutations are resistant to PD-1 or PD-L1 monoclonal antibodies [30]. (See KRAS mutations below). Interestingly, β2-adrenergic receptor (β2-AR) is activated by neurotransmitters, such as, norepinephrine, in EGFR mutant NSCLC. Activation of β2-AR inactivates LKB1, with upregulation of CREB and interleukin-6 (IL6) [54]. Propranalol (β2-AR inhibitor) blocks norepinephrine induced IL6. These findings pave the way for further improvement of the efficacy of anti-PD-1 and anti-PD-L1 antibodies, with the aid of repurposing drugs, such as, gefitinib or propranolol (Table 2). Interestingly, a phase Ib/II study (NCT03384836) of propranolol with pembrolizumab is ongoing in patients with unresectable stage III and IV melanoma.

Neurotransmitters have been shown to correlate with the formation of myeloid-derived suppressor cells (MDSCs) (see below). The microbiome has been postulated to influence the response to immunotherapy and, interestingly, has been associated with cardiovascular disease (myocardial infarction and stroke). A gut microbiota-derived metabolite, phenylacetylglutamine (PAGln) was seen to increase the platelet activation and thrombosis. PAGln mediates cellular events through several adrenergic receptors. The findings support the clinical benefit of better blocker therapy, may be in part mediated by attenuation in PAGln-triggered adrenergic receptor (ADR) signaling. In previous studies, it has been seen that carvedilol promotes inhibition of platelet function [31].

IL6-STAT3 signaling is noteworthy since it causes immunosuppressive features in NSCLC. A recent study shows that IHC staining with overexpression of STAT3 correlates with repressed infiltration of CD8+ T cells in NSCLC and augmenting the number of MDSCs. Numerous studies indicate that targeting STAT3 could be an ancillary therapy to augment the effect of immunotherapy (Table 2). Niclosamide has been shown to augment the effect of PD-L1 monoclonal antibodies by blocking STAT3 activation [55].

Although pyroptosis was not found to be involved in the effect of gefitinib in ID1 overexpressing NSCLC cell lines [28], it has been proven that MLKL signaling activates the NOD-like receptor protein 3 (NLRP3) inflammasome, triggering caspase-1 processing of the pro-inflammatory cytokine, IL-1β. However, GSDMD, the pore-forming caspase-1 substrate required for efficient NLRP3-triggered pyroptosis and IL-1β release, is not essential for MLKL-dependent death or IL-1β secretion [32]. Bioinformatic analysis has shown that IL-1β is associated with EGFR-TKI resistance in NSCLC. IL-1β upregulates C-terminal Eps15-homology (EH) domain-containing protein (EHD1). EHD1 regulates multiple steps of endocytosis and vesicle trafficking, regulating microtubule dynamics [56]. It was previously noted that EHD1 contributes to erlotinib resistance in EGFR-mutant NSCLC. A model was established for a regulatory signaling network of NF-ĸB/mIR-590/EHD1, in which NF-ĸB suppressed the expression of mIR-590 and increased the expression of EHD1 [57]. Upregulation of EHD1 is also associated with metastasis in NSCLC [58].

Caspase-8 is the initiator caspase of extrinsic apoptosis, and, also, connects via BID with intrinsic apoptosis pathway [59]. Caspase-8 also inhibits necroptosis mediated by RIPK3 and MLKL [60]. More recently, caspase-8 has been shown to be a molecular switch controlling apoptosis, necroptosis and pyroptosis. Caspase-8 is involved in the activation of the infammasome and induction of pyroptosis, under circumstances where apoptosis and necroptosis are compromised. Viruses are reliant on the fate of infected cells and are suppressors of apoptosis and necroptosis. Therefore, it is speculated that viral inhibitors can activate pyroptosis, as a host defense [61]. Ferroptosis is a form of regulated cell death that is caused by the iron-dependent peroxidation of lipid. Glutathione peroxidase 4 (GPX4) prevents ferroptosis by converting lipid hydroperoxides into non-toxic lipid alcohols. However, GPX4 inhibitors do not always cause cell death in cancer cell lines. In parallel to the canonical GPX4 pathway, the ferroptosis suppressor protein (FSP1) positively correlates with ferroptosis resistance across hundreds of cancer cell lines and FSP1 mediates resistance to ferroptosis in lung cancer cells in culture and mouse tumor xenografts [62].

## 5. NLRP3 and IL-1β

Accumulated DNA in the cytosol is a central immunostimulatory signal associated with infection, cancer and genomic damage. Cytosolic DNA triggers immune response by activating the cyclic GMP-AMP (cGAMP) synthase (cGAS)-STING pathway. The binding of DNA to cGAS activates its enzymatic activity, leading to the synthesis of a second messenger, cyclic guanosine monophosphate-adenosine monophosphate (2′3′-cGAMP). cGAMP binds to STING and triggers its translocation from the endoplasmic reticulum to the Golgi intermediate compartment (ERGIC) and Golgi. The cGAMP-bound STING can translocate through the trans-Golgi network and endosomes to lysosomes for degradation via the multi-vesicular body (MVB) pathway [63]. Furthermore, it has been shown that cGAMP induces autophagy for the clearance of DNA in viruses in the cytosol. The DNA inflammasome initiated by cGAS-cGAMP-STING leads to the lysosomal rupture with potassium efflux, in turn, leading to NLRP3 inflammasome activation, releasing IL-1β [64]. Overexpression of IL-1β induces gastric cancer and mobilizes MDSCs in mice [33,34]. Moreover, IL-1β inhibition with canakinumab reduces the incidence of lung cancer in patients with atherosclerosis [35]. Blocking IL-1β reverts the immunosuppression in mouse breast cancer and enhances the activity of anti-PD-1 monoclonal antibodies (Table 2) [36]. Interleukin-1β enhances tumor development by contributing to chronic inflammation, tumor angiogenesis and metastasis, and induction of immunosuppressive cells in the tumor microenvironment. However, due to its key role as a downstream mediator of inflammation, IL-1β has been shown to have tumor-promoting, as well as anti-tumorigenic effects. Indeed, IL-1β induces antigen-specific T cell responses by driving polarization of CD4+ T cells towards T helper type (Th) 1 and Th17 cells and enhances adaptive anti-tumor responses. Hence, its role in predicting response to immune checkpoint blockade needs to be further evaluated, and further research is needed to assess the efficacy of IL-1β inhibitors with immunotherapy or other anti-cancer therapeutic agents [37]. Of note, an ongoing randomized, phase 2 trial (NCT03968419) is evaluating the IL-1β inhibitor, canakinumab, or pembrolizumab, as monotherapy or in combination, as neoadjuvant therapy in patients with resectable NSCLC, with major pathological response (MPR) rate as the primary endpoint.

Moreover, the NLRP3 inflammasome is regulated by the secretion of bile acids. Bile acids activate the cell membrane receptor TGR5 (transmembrane G protein-coupled receptor-5), able to inhibit NLRP3 inflammasome activation. TGR5 activates cAMP-PKA kinase and induces NLRP3 phosphorylation on Ser291 and ubiquitinates NLRP3. In short, TGR5 signaling prevents metabolic disorder by inhibiting NLRP3 inflammasome. The findings indicate that TGR5 can also provide information on the effect of anti-PD-1 and anti-PD-L1 monoclonal antibodies. The stress granule protein DDX3X (DEAD-Box Helicase 3 X-Linked) regulates inflammasome activation. The function of DDX3X warrants investigation, as it can either function in pyroptosis, or as a pro-survival signal. NLRP3 can be driven by TLRs-NFĸB signaling and artemisinin has been considered an inhibitor of NFĸB [27,65].

Dihydroartemisinin was previously shown to be a STAT3 inhibitor [27,66] STAT3 is activated in mesenchymal cells, leading to promotion of DNA methyltransferases (DNMTs), which further methylate the promoter region of key molecules in antigen presentation machinery, such as, IRF1, PSMB8, PSMB9 and HLA molecules. STAT3 can further inhibit expression of STAT1, a key regulator of antigen presentation machinery in epithelial cells [67]. cGAS-STING signals type I IFN response, upstream and independent of NLRP3. We observed that high IFN*γ* mRNA levels were an important marker of response to pembrolizumab in melanoma patients, and also an indicator of longer survival in NSCLC patients treated with nivolumab. However, the levels of PD-L1, STAT1, retinoic acid inducible gene 1 (RIG1), YAP1, RANTES, and other transcripts, were not associated with progression-free, nor overall survival [68]. However, the overexpression of IFNγ could be nefarious and it has been seen that aspirin limits the cGAS activation by acetylation [69]. The Toll-like receptor 4 (TLR4) is a specific sensor of exogenous microbial ligands, such as, lipopolysaccharides, as well as, damage-associated molecular patterns (DAMPs) derived from host tissue or cells. Pancreatic microenvironment is abundant in TLR4 ligands, including HMGB1 and S100A9, that can activate TLR4 signaling in tumor cells [70]. A fungus called Malassezia promotes pancreatic ductal adenocarcinoma [71].

In the study by Pfirschke et al., immunogenic chemotherapy, such as oxaliplatin and cyclophosphamide, induced tumor cell release of HMGB1, and activation of TLR4 on DCs that, in turn, stimulate antitumor CD8 + T cells. These results suggest a role for some drugs in sensitizing tumors to immune checkpoint therapy [72].

## 6. CFTR and Mucins

In muco-obstructive lung diseases a muco-inflammatory pathway has been identified, in which activated resident macrophages release IL-1β and hypoxic necrotic epithelia release IL-1α, then IL-1α and IL-1β activate epithelial IL-1 receptors to induce mucin biosynthesis. The effects in the secretion of chloride and bicarbonate anions mediated by the cystic fibrosis transmembrane conductance regulator (CFTR) are present in cystic fibrosis [73]. The airway epithelial defects produce mucus hyper-concentration in chronic obstructive pulmonary disease (COPD). Exposure to cigarette smoke induces abnormalities in CFTR-mediated secretion of chloride anions through oxidant-induced reduction of CFTR transcription rates and direct damage to CFTR protein in the apical membrane. The effects in the epithelial ion and fluid transport (hydration) are amplified by cigarette smoke-induced hyper-secretion of MUC5AC and MUC5B mucins [74]. Looking at the CFTR levels in lung cancer patients, it could be of great interest to further decipher the contribution of the mucins in response to anti-PD-1 or anti-PD-L1 monoclonal antibodies. It is tempting to speculate that treatment with CFTR modulators could be of benefit in lung cancer patients with COPD. Therapy with elexacaftor-tezacaftor-ivacaftor for cystic fibrosis with CFTR mutations has provided benefit [75]. MUC1-C represses RAS association domain family IA (RASSF1A) expression and KRAS wild-type and mutant NSCLC. MUC1-C is an oncoprotein that associates with RTKIs in the cell membrane, promoting the activation of their downstream signaling pathways, including PI3K/AKT and MEK/ERK. Of note, MUC1-C increased PD-L1 transcription through MYC- and NF-κB p65-mediated mechanisms in triple negative breast cancer (TNBC) cells, supporting its involvement in immune evasion [76]. Similar mechanisms of PD-L1 expression induced by MUC1-C have been reported in NSCLC cells [77]. Targeting MUC1-C in vitro and in vivo showed downregulation of PD-L1 expression by TNBC cells and activation of the tumor immune microenvironment through an increase in tumor infiltrating CD8+ T-cells [76]. Therefore, the potential relationship of MUC1-C alone or with COPD and CFTR is particularly of interest regarding the response to anti-PD-1 and anti-PD-L1 antibodies.

## 7. Neoadjuvant Anti-PD-1 or Anti-PD-L1 Monoclonal Antibodies

It has been considered that immunotherapy could be more active if used as neoadjuvant (pre-surgical) therapy in early cancer, including NSCLC. The reasons include, that before removal of tumor-draining lymph nodes (TDLN), the activity of anti-PD-1 or anti-PD-L1 monoclonal antibodies could be crucial, since lymph nodes are essential for anti-PD-1 activity, where dendritic cell presentation of tumor antigens to T cells is enhanced [1,78]. Several points will require clarification in forthcoming clinical trials, such as the sequential time between neoadjuvant immunotherapy and time of performing surgery. It has been demonstrated that a short interval between first administration of immunotherapy and resection of the primary tumor is important for optimal efficacy, while extending the duration abrogated immunotherapy efficacy [79]. The neoadjuvant studies look to increase the rate of MPR, resulting in less than 10% of residual viable tumor (RVT) [1]. Notwithstanding, pathologic complete response is the primary goal of neoadjuvant therapy. In the future, defining immune mediated tumor regression criteria should be further clarified. For example, the percent of RVT is calculated by the surface area of the RVT-surface area of the tumor bed. The tumor bed surface area includes, RVT, plus tumor-associated stroma, plus necrosis, plus regression bed [1]. Nivolumab (PD-1 monoclonal antibody) was administered at a dose of 3 mg per kilogram of body weight every 2 weeks with surgery planned 4 weeks after the first dose in surgically resectable early (stage I, II, or IIIA) NSCLCs. MPR occurred in 45% of patients. Responses were seen in both PD-L1 positive and PD-L1 negative tumors [80]. Neoadjuvant atezolizumab (anti-PD-L1 antibody, 1200 mg) on day 1, nab-paclitaxel (100 mg/m^2^) on days 1, 8 and 15, and carboplatin (AUC 5) on day 1, every 21 days, was administered up to four cycles in surgically resectable stage IB-IIIA patients. MPR was observed in 57% of patients and pathologic complete response in 33% of patients [81]. The results of these two neoadjuvant studies indicate that chemotherapy alone or in combination with immunotherapy still has limited efficacy and multiple biological variables should be kept in mind for augmenting the efficacy of ant-PD-1 or anti-PD-L1 monoclonal antibodies. Some clues have been described above. Similarly, neoadjuvant immunotherapy has been employed in other primary tumors, including melanoma [1]. Triple negative breast cancer (TNBC) shares some commonalities with NSCLC, including upregulation of AXL. Pembrolizumab (200 mg) every 3 weeks, plus paclitaxel and carboplatin was given as a neoadjuvant approach in comparison with chemotherapy in early TNBC. Pathological complete response was observed in 64.8% in the pembrolizumab−chemotherapy group and 51.2% in the chemotherapy group (*p* < 0.001). Benefit was observed among the majority of prognostic risk categories, including patients with low PD-L1 expression [82]. Notwithstanding, the room for improvement in PFS and OS is still large. In advanced TNBC, atezolizumab plus nab-paclitaxel yields a PFS of 7.2 months, in comparison with nab-paclitaxel of 5.5 months [83]. In advanced NSCLC, as commented in the beginning, the survival outcomes in different studies is increased in comparison with chemotherapy, but still requires further measures in order to allow anti-PD-1 or anti-PD-L1 monoclonal antibodies to be more efficacious. A cooperative study carried out in Latin America shows that the use of immunotherapy in first line has a better median overall survival than when used in second or third line [84]. New approaches are promising, such as the use of adjuvant epigenetic therapy with low dose DNA methyltransferase and histone deacetylase inhibitors, 5-azacytidine and entinostat, which disrupt the premetastatic niche by inhibiting the trafficking of MDSCs through downregulation of CCR2 and CXCR2. Such therapy decreases the accumulation of MDSCs in the premetastatic lung, producing longer periods of disease-free survival and increased overall survival in comparison with chemotherapy [85]. (For further information on CCR2 and CXR2, see [68]). Another unexplored target to increase the effectiveness of immunotherapy is the relationship of Src homology 2 (SH2) domain containing phosphatase 2 (SHP2) activation in the PD-1 pathway [86]. SHP2 is a critical regulator of RAS MAPK pathway and SHP2 inhibitors have activity in cancers with KRAS mutations and BRAF mutations [87]. SHP2 inhibitors could target cancer cells, T lymphocytes, and macrophages simultaneously, and, thereby, could have significant activity in conjunction with anti-PD-1 and anti-PD-L1 antibodies (Table 2) [88].

## 8. Post-Translational Modifications of PD-L1: N-Linked Glycosylation of PD-L1 as Predictive Biomarker and Therapeutic Target

Glycosylation is a post-translational modification of proteins and is involved in numerous processes in cancer cells. There are two types of protein glycosylation, N-linked and O-linked. The synthesis of N-glycans is initiated in the endoplasmic reticulum (ER) by transfer of a preformed lipid (dolichopyrophosphate)-linked oligosaccharide precursor containing three glucoses, nine mannoses, and two-N-acetylglucosamines to asparagine of nascent proteins that have entered the ER lumen [38]. N-glycans are divided into three types, according to the sugar moiety structures: high-mannose type, hybrid type, and complex type. PD-L1 is highly glycosylated with heterogenous expression patterns on western blots. Although the 33-kDa form of non-glycosylated PD-L1 can be detected, the majority of PD-L1 is glycosylated with heterogenous molecular weight, ranging from 45-55 kDa on western blots [89]. Treatment with recombinant glycosidase (peptide-N-glycosidase F; PNGase F) removed the N-glycan structure and a significant portion of the 45-kDa PD-L1 reduced to 33 kDa. The addition of recombinant glycosidase, endoglycosidase H (Endo H) which cleaves high-mannose and some hybrid oligosaccharides, only partially reduced PD-L1 glycosylation, indicating that the complex-type of N-linked glycan structures exists predominantly on PD-L1 (Figure 1). It is tempting to speculate that PD-L1 glycosylation could be a major mechanism of resistance to anti-PD-1 and anti-PD-L1 antibodies. PD-L1 glycosylation can be inhibited by tunicamycin (N-linked glycosylation inhibitor) [89]. Glycogen synthase kinase 3β (GSK3β), through β-TrCP, leads to the ubiquitination of PD-L1. EGF signaling induces PD-L1 glycosylation, by inhibiting GSK3β (phosphorylation of Ser9). Other EGFR ligands, such as, epiregulin, TGFα, and heparin-binding EGF, also induce PD-L1 expression (Figure 1). Unlike IFNγ-induced PD-L1 via transcription activation, the action of EGF-mediated PD-L1 induction is primarily at post-translational level. The observation suggests that activation of EGFR inactivates GSK3β, stabilizing PD-L1 expression (Figure 1) [89]. Inhibition of EGF-mediated PD-L1 stabilization enhances the efficacy of PD-1 blockade, promoting tumor-infiltrating cytotoxic T cell immune response [89]. A recent report suggests that listeria-based hepatocellular cancer vaccine facilitates anti-PD-1 therapy by regulating macrophage polarization [90]. In addition, a PD-L1-targeted vaccine shows strong anti-tumor activity [91]. It is tempting to speculate that the addition of EGF vaccine to anti-PD-1/PD-L1 antibodies could enhance the efficacy in several classes of NSCLC. In our experience, EGF vaccine antibodies reduce the expression of EGFR (pEGFR Y1068), as well as the expression of EGFR ligands [92]. Intriguingly, several mechanisms inducing N-linked glycosylation of PD-L1 have been reported [93]. Glycosylated PD-L1 expression has also been observed in antigen presenting cells (APCs), like dendritic cells and macrophages. PD-L1 protein is heavily glycosylated in TNBC cells and quantitative qPCR analysis shows that β-1,3-N-acetylglucosaminyl transferase (B3GNT3) was upregulated by EGF in TNBC cell lines. A strong correlation between B3GNT3 and EGFR was noted. Breast and lung cancer patients who had high B3GNT3 expression have poor overall survival. Chromatin immunoprecipitation-sequencing data revealed that TCF4 downstream of the EGF-GSK3β-β-catenin pathway bound directly to the B3GNT3 core promoter region [94]. EMT induces N-glycosyltransferase STT3 through β-catenin (Figure 1). STT3 positively regulates PD-L1 glycosylation, leading to PD-L1 induction in cancer stem cells and non-cancer stem cells. Etoposide, a cytotoxic drug, reduces nuclear β-catenin through TOPB2 degradation and inhibits the EMT-β-catenin-STT3-PD-L1 signaling [95]. Multiple other mechanisms influence PD-L1, such as, the deubiquitination and stabilization of PD-L1 by CSN5. Of interest is the fact that curcumin acts as a CSN5 inhibitor [96]. CKLF-like MARVEL transmembrane domain containing protein 6 (CMTM6) is a major regulator of PD-L1 expression in cell lines of melanoma, breast and lung cancer. CMTM6 depletion reduces both constitutive and IFNγ-induced PD-L1 expression. CMTM6 is a master regulator of PD-L1 cell surface expression. CMTM6 associates with PD-L1 at the plasma membrane, and, in recycling endosomes, protects PD-L1 from being targeted for lysosomal degradation [97]. Other studies have also demonstrated that CMTM6 is present at the cell surface, associates with the PD-L1 protein, reduces its ubiquitination, and increases PD-L1 protein half-life [93,98]. Recently, it has been observed that enzymatic deglycosylation of tissue sections improves PD-L1 detection in tumor samples. It was stated that deglycosylated PD-L1 is a more reliable biomarker to predict immunotherapy response [99]. However, the technique is not so easily reproducible, and the interpretation requires further validation, since PD-L1 glycosylation could be a mechanism of resistance to anti-PD-1/PD-L1 monoclonal antibodies. Further insights on the involvement of the Golgi should be explored. For example, it has been reported that mutated KIT in GIST cell lines accumulated on the trans-side of the Golgi apparatus, where mutated KIT is phosphorylated on Y703, but not on the ER. The effects of HSP90 inhibitors on KIT localization and phosphorylation were determined using confocal microscopy with GM130 as a Golgi marker and PDI as an ER marker. Treatment with HSP90 inhibitors decreased Golgi-localized KIT. KIT phosphorylated on Y703 almost completely disappeared from the Golgi apparatus, following the addition of HSP90 inhibitors [100]. The lower and upper KIT bands are high-mannose form (HM, endo H-sensitive, ER and cis-side of Golgi) and complex-glycosylated form (CG, endo H-resistant, reaching trans-side of Golgi), respectively. As KIT is fully activated in the Golgi in the CG form, HSP90 inhibitors predominantly decrease KIT on the trans-Golgi compared with that on the ER and cis-Golgi [100].

## 9. Glutathione Synthesis and Ferroptosis: Cystine-Glutamate Antiporter (xCT)-GSH/GPX Pathway and p53

NSCLC, like other tumors (TNBC, for example), has increased dependence, not only on glucose, but on cystine or glutamine as well. Smoking induces the expression of xCT (SLC7A11) in NSCLC cells. xCT (SLC7A11) is a cystine/glutamate antiporter that imports cystine into the cells, while exporting glutamate (Figure 2). One molecule of cystine can then be converted into two molecules of cystine, a necessary step for glutathione (GSH) biosynthesis [39,40,101]. To quench reactive oxygen species (ROS), GSH is oxidized to GSH disulfide (GSSG), a reaction requiring nicotinamide adenine dinucleotide phosphate (NADPH). Such mesenchymal therapy-resistant cells are dependent on the lipid hydroperoxidase GPX4 for survival [41] (Figure 2). Sulfasalazine (SASP), a FDA-approved drug, has inhibitory effects on the function of xCT, by decreasing the supply of cystine [101]. xCT is overexpressed in NSCLC and correlates with worse survival. xCT is highly expressed in different NSCLCs, such as A549, H520, and others, but not, for example, in H1869. Cell proliferation assays show dose-dependent growth induction induced by SASP in A549 and H520, but not in H1869.To maintain the intracellular glutamate pool, cells overexpressing xCT consume more glutamine for glutamate synthesis, a process of glutamine addiction [101]. Metabolic reprogramming by elevated glucose consumption, lactate production, and glutamine addiction, is a feature in NSCLC. After 72 h treatment, cigarette smoking condensate increases the expression of xCT in BEAS2B cells (normal airway epithelial cells).

It has been shown that immunotherapy-activated CD8^+^ T cells enhance ferroptosis-specific lipid peroxidation in tumor cells, and that increased ferroptosis increases the anti-tumor efficacy of immunotherapy [102]. Intriguingly, IFNγ released from CD8^+^ T cells downregulates the expression of SLC7A11 and of SLC3A2, another subunit of the glutamate-cystine antiporter system (xCT), henceforth, limiting the uptake of cystine by tumor cells, promoting tumor cell lipid peroxidation and ferroptosis [102] (Figure 2). The benefit of nivolumab correlated with reduced expression of SLC3A2 and increased IFNγ and CD8+. This study highlights that T cells-promote tumor ferroptosis and that targeting this pathway (xCT-GSH/GPX) in combination with anti-PD-1 or anti-PD-L1 antibodies could be a useful strategy (Table 2). Reinforcing the relevance of xCT, the combination of tumor cell xCT deletion with anti-CTLA, increased the frequency and durability of anti-tumor response [103]. Several compounds can induce xCT disruption and potentially augment the immunotherapy efficacy [42] (Figure 2). Intriguingly, sorafenib, a multi-kinase inhibitor currently approved by EMA and FDA for hepatocellular carcinoma, advanced renal cell carcinoma, and thyroid carcinoma, is a xCT inhibitor and can deplete GSH and induce ferroptosis. Dihydroartemisinin (DHA) is a derivative and active metabolite of artemisinin, decreasing the levels of glutathione peroxidase 4 (GPX4) in head and neck squamous cell carcinoma (HNSCC) cells [43]. DHA also blocks the forkhead box protein M (FOXM1), a transcriptional master regulator. We have noted that DHA can inhibit important downstream signaling nodes, such as, STAT3, YAP1, AKT, MEK, and several RTKs, including MET [27]. A feedback loop between FOXM1 and MET has already been discovered [44].

p53-mediates ferroptosis by repressing SLC7A11 transcription [104], indicating that p53 status is important for predicting the effect of anti-PD-1 and/or anti-PD-L1 antibodies in gauging the cystine/glutamate xCT antiporter system components, as potential therapeutic targets for SASP, DHA, or other drugs [39] (Figure 2). The sensitivity of ROS-induced ferroptosis is increased in p53-activated cells. The levels of SLC7A11 are critical for the sensitivity of ferroptopic responses [104]. In addition, the p53-related lncRNA (P53RRA) is also involved in the regulation of SLC7A11. Increased expression of P53RRA enhances its interaction with ras-GTPase-activating protein-binding protein 1 (G3BP1) (Figure 2). GRBP1 modulates the transduction of signaling stimulated by the oncoprotein Ras. P53RRA sequesters more p53 in the nucleus after P53RRA binds to the GRBP1 RRM binding motif. P53RRA regulates metabolic genes, including SLC7A11 and TIGAR (p53-inducible regulator of glycolysis and apoptosis), promoting the accumulation of lipid ROS and intracellular iron, then inducing ferroptosis and apoptosis and cell cycle arrest in a p53-dependent manner [105]. A549 is a cell line that highly expresses P53RRA [105], however, this is in contradiction with the other study above mentioned where A549 displays high levels of SLC7A11 [101]. It was stated that P53RRA acts upstream of the p53 signaling pathway [105]. The chromatin remodeling factor lymphoid-specific helicase (LSH) silences P53RRA directly [105]. In addition, the chromatin remodeling factor lymphoid-specific helicase (LSH) activates proline dehydrogenase (PRODH), which catalyzes proline to yield pyrroline-5-carboxylic acid (P5C). PRODH promotes NSCLC by inducing EMT and IKKα-dependent inflammatory genes. LSH and PRODH mRNA levels are increased in lung adenocarcinoma using TCGA database. Using the compound L-tetrahydro-2-furoic acid (L-THFA) inhibits PRODH [106]. The function of PRODH could be clinically relevant in NSCLC with wild-type TP53.PRODH promotes tumor cell survival through, either, ATP production, ROS generation, or autophagy induction, when cells are under nutrient limited conditions [106]. Other studies show paradoxical findings, such as the fact that, increased ROS following TIGAR or nuclear factor erythroid-2 related factor 2 (NRF2) loss enhances metastasis in pancreatic cancer [107].

Glutamine-avid TNBC (one-third of TNBCs) is sensitive to SASP. Other compounds targeted SLC7A11 expression and cystine/glutamate exchange activity, including an inhibitor of aminotransferases (6-diazo-5-oxo-L-norleucine; DON), or prevented glutamine access with asparaginase, reducing asparagine and glutamine to their acidic derivatives [39] (Figure 2). Asparagine availability in vitro strongly influences EMT and could regulate metastatic progression [108]. Limiting asparagine by knockdown of asparagine synthetase (ASNS) treatment with L-asparaginase, or dietary asparagine restriction, reduces metastases [108]. Suppression of ASNS increases the potency of artemisinin in an NSCLC model in which artemisinin activates the ER-resident PERK, which reactivates ERK, phosphorylating ATF4, and nuclear ATF4 can enhance ASNS production [109]. Phosphorylation of ATF4 is present in BRAF plus MEK inhibitor-resistant tumors via GRP78/KSR2/PERK [110]. Noteworthy is the fact that KRAS mediates the production of ATF4, especially in the presence of KEAP inactivating mutations, which leads to upregulation of NRF2 and more ATF4 production, which leads to increased intracellular ASN. The combination of L-asparaginase for asparagine depletion, plus AKT inhibition, has been suggested for treatment of NSCLC with KRAS mutations [111].

## 10. Metabolic Rewiring

Aberrant lipid metabolism is observed in cancer, which can, therefore, modulate the effect of IMT. Fatty acid synthase (FASN) is a critical enzyme for the synthesis of long-chain fatty acids from malonyl-CoA. FASN is upregulated in several cancer types. FASN overexpression in tumors is dependent on the PIEK-AKT signal transduction pathway and SREBP 1c transcriptional regulation. In EGFR mutant NSCLC cells, resistance to gefitinib is related to FASN, which produces 16-C saturated fatty acid palmitate (palmitoylation). The schema points out that gefitinib-untargetable palmitoylated EGFR impinges on FASN, promoting tumor growth via the AKT pathway. Pharmacological inhibition of FASN by orlistat (Xenical) (an FDA-approved anti-obesity drug) inhibits EGFR palmitoylation, limiting tumor cell survival [112]. Signal-transducer and activator of transcription (STAT3), which plays a central role in IMT, can be post-translationally S-palmitoylated through ZDHHC19, a palmitoyl acyltransferase. ZDHHC19 is often amplified in multiple cancers, especially in lung squamous cell carcinoma (39%) [113]. Palmitoylation of KRAS4A (a KRAS product) allows affinity for the plasma membrane. After depalmitoylation, KRAS4A loses affinity for the plasma membrane and gains affinity for endomembranes. Tethering of KRAS4A to the outer mitochondrial membrane allows it to interact with hexokinase 1 (HK1, the initiator of glucose metabolism). Such interaction activates glucose-6-phosphate and enhances HK1 activity and glycolytic flux [114]. Therefore, KRAS4A palmitoylation inhibition could be a supplemental treatment for enhancing the effect of anti-PD1 or anti-PD-L1 antibodies.

Inhibiting cholesterol esterification in T cells by ablation or pharmacological inhibition of acetyl-CoA Acetyltransferase (ACAT1) augments the effector function and proliferation of CD8^+^ cells. The effect is due to the increase in the plasma membrane cholesterol level of CD8^+^ T cells. Avasimibe (used to treat atherosclerosis) was used as an ACAT1 inhibitor. The combination of avasimibe with anti-PD-1 antibody inhibits tumor progression in mice models. Avasimibe monotherapy potentiates the effector function of both, PD-1^high^, and PD-1^low^, CD8^+^ T cells in the tumor microenvironment [115]. ACAT1 inhibition could be complementary to IMT (Table 2). Cancer cells can promote membrane-cholesterol efflux and depletion of lipid rafts from macrophages. Increased cholesterol efflux promotes IL-4-mediated reprogramming, involving inhibition of IFNγ-induced gene expression. Targeting ABC transporters that mediate cholesterol efflux could also be of therapeutic interest [45]. ACAT1 is upregulated in response to extra palmitic acid (PA). It is enticing to know that ACAT1 mediates glycerophosphate O-acyltransferase (GNPAT) acetylation, which stabilizes FASN to facilitate lipid metabolism and hepatocarcinogenesis [116]. Combination therapy with sorafenib and an ACAT1 inhibitor attenuates tumor growth in the xenograft hepatocellular model. Moreover, in primary liver and lung cancer cell lines, it is noted that desaturate palmitate (fatty acid desaturation) to the unusual fatty acid sapienate, induces proliferation. Sapienate biosynthesis permits cancer cells to bypass the fatty acid desaturation pathway, placing sapienate biosynthesis as an alternative source of monounsaturated fatty acids [117].

Type I IFNs are induced following activation of cell-surface or intracellular pattern recognition receptors (PRRs), such as, retinoic-acid-inducible gene I (RIG-I)-like receptors (RLRs), STING and TLRs [68,118,119]. RLR triggers mitochondrial antiviral-signaling (MAVS). RIG-I prevents hexokinase 2 (HK2) binding to MAVS. Lactate is responsible for glycolysis-mediated RLR signaling inhibition and its inhibition increases type I IFN production. The study identifies MAVS as a sensor of lactate connecting energy metabolism and innate immunity. Treatment with sodium oxamate (a specific LDHA inhibitor) enhances IFNβ production [120].

Polymorphonuclear myeloid-derived suppressor cells (PMN-MDSCs) are pathologically activated neutrophils that abrogate the IMT efficacy. Of great interest is the fact that PMN-MDSCs exclusively upregulate fatty acid transport protein 2 (FATP2). Lipofermata inhibits FATP2 in tumor-bearing mice. Treatment of LLC-bearing mice with the combination of anti-CTLA4 antibody and lipofermata had a potent antitumor effect with four out of five mice rejecting tumors. The main mechanism of FATP2-mediated suppressive activity implies the uptake of arachidonic acid and the synthesis of prostanglandin E_2_ [121].

## 11. Immunotransmitters: the cGAS-STING Pathway

As commented above, cGAMP plays a central role in anticancer and antiviral innate immunity. It is synthesized by the enxyme cyclic GMP-AMP synthase (cGAS) in response to double-stranded DNA in the cytosol. cGAMP binds and activates its endoplasmic reticulum (ER) surface receptor STING to activate production of type I IFNs. cGAMP binds to STING and triggers STING translocation from the ER to the ERGIC and Golgi (Figure 3). cGAMP bound STING translocates through the trans-Golgi network and endosomes to release TANK-binding kinase 1 (TBK1)-interferon regulatory transcription factor 3 (IRF3) and production of IFNs [63], as previously mentioned. Moreover, STING can activate NF-κB pathway by binding to IκB kinase (IKK) and NF-κB-inducing kinase (NIK), collaborating with TBK1-IRF3 pathway to induce the expression of type I IFN (Figure 3) and exert its multiple immune-stimulatory functions. New evidence shows that cGAMP is also exported to the extracellular space to signal other cells. However, the ectonucleotide pyrophosphatase phosphodiesterase 1 (ENPP1), a cGAMP hydrolase, blocks the export of cGAMP. Inhibitors of ENNP1 can boost extracellular cGAMP concentration, immune infiltration and synergize with irradiation to delay tumor progression [122]. Irradiation induces more than a tenfold increase in extracellular cGAMP levels in E0771 cells, than in other cell lines. In E0771 orthotopic tumors in mice, neutralizing STING (to deplete extracellular cGAMP) significantly decreases the CD11c^+^ (dendritic cell) and CD103^+^CD11c^+^ (conventional type 1 dendritic cell, CDC1) populations. The experiments reveal that extracellular cGAMP can be detected by the immune system to activate dendritic cells that are important for the anticancer response [122]. ENNP1 is elevated in some breast cancers and related to poor prognosis. 

High ENPP1 expression in breast cancer depletes extracellular cGAMP dampening immune detection. In short, cancer cells produce soluble extracellular cGAMP as a danger signal, increasing the number of dendritic cells and cytotoxic T cell activation in the tumor microenvironment. cGAMP export permits cGAMP communication among cells in close proximity. It is proposed that cGAMP is an immunotransmitter sharing properties with neurotransmitters in immune signaling functions [122]. Therefore, extracellular (soluble cGAMP) and tumor ENPP1 expression could help to predict the benefit of IMT in stage III, where durvalumab has shown a survival improvement after chemoradiotherapy [123].

The same investigators [124], and others [125], have identified SLC19A1 as a cGAMP importer. The SLC19A1 protein, aka, Reduced Folate Carrier (RFC1), was previously characterized as a high-affinity importer of reduced folates, such as folinic acid, and of antifolates, such as methotrexate. The results indicate that not only methotrexate, but also sulfasalazine, can inhibit cGAMP, and the uptake of other cyclic dinucleotides (CDNs) in human cell lines, while SLC9A1 overexpression increases both uptake and functional responses [125]. Therefore, SLC19A1 expression could predict the STING function that can be impaired in combination with IMT and antifolate drugs. Therefore, both endogenous and exogenous extracellular cGAMP promotes immune cell recruitment and tumor shrinkage in a STING-dependent manner. SLC19A1 is the dominant cGAMP importer in monocyte-derived U937 cells (derived from a human histiocytic lymphoma), as well as in THP-1 (AML) cells. Genetic knockout of SLC19A1, and inhibition of import with methotrexate, reduced pIRF3 signal by 50% in response to extracellular cGAMP. In addition to CGAMP, bacterial CDN second messengers, including 3′3′-cyclic-GMP-AMP (3′3′-cGAMP) and other CDNs, also bind and activate the STING pathway [124,126]. Due to its key role in regulating anti-cancer immune responses, positive modulation of the cGAS-STING pathway signaling has been proposed as a potential therapeutic strategy to enhance tumor immunogenicity and sensitivity to a variety of immunotherapies, such as, cancer vaccine and immune checkpoint blockade [127]. In several preclinical cancer models, pharmacological activation of STING has been shown to restrict tumor growth and enhance immunogenicity. Upregulation of IFN by cGAS-STING can contribute to the increase of antigen presentation molecules, including major histocompatibility complex (MHC) and to the activation of antitumor killing activity of cytotoxic T lymphocytes. STING agonists promote CD8+ tumor infiltration, a precondition of improved response to anti-PD-1/PD-L1 treatment, and induce PD-L1 expression on tumors [127]. The combination of 2′3′-cGAMP treatment with PD-L1 inhibitor has shown promising results in a xenograft model [46]. STING agonists, such as ADU-S100, MK-1454 and GSK3745417, are currently being tested in tumors in phase 1 and 2 trials, including in combination with immune checkpoint inhibitors [47].

Cancer cells have unstable genomes that result in improper chromosome segregation during mitosis. This leads to the formation of micronuclei enclosed by leaky membranes, thereby exposing dsDNA to the cytosol and activating the cGAMP-STING pathway [46,124,126]. cGAS also promotes cancer progression by inhibiting DNA repair [48] and may also rewire the STING pathway to promote metastasis, while avoiding IFNβ production [46]. Recognition of ruptured micronuclei (DNA damage) by cGAS induces nuclear translocation of cGAS dependent on importin-a, and the phosphorylation of CcGAS at tyrosine 215 by B-lymphoid tyrosine kinase (BLK). In the nucleus, cGAS is recruited to double-stranded breaks and interacts with PARP1. The cGAS-PARP1 interaction impedes the formation of the PARP1-Timeless complex, suppressing homologous recombination [48]. The phosphorylation of cGAS at Y215 by BLK maintains the cytosolic localization of cGAS, while stimulation with DNA damaging agents dephosphorylates cGAS shuttling cGAS to the nucleus. In summary, cGAS acts as a tumor enhancer. Aberrant upregulation of cGAS transcripts is observed in NSCLC, including squamous cell carcinoma and adenocarcinoma. cGAS interact with proteins of the importin-a family, including karyopherin (KPNA2), and others. KPNA2^high^BLK^low^ expression levels are associated with poor prognosis in NSCLC. Inhibitors that interfere in the nuclear translocation of cGAS have been suggested as adjuncts to chemotherapy and irradiation [48].

## 12. Neurotransmitters and Associated Receptors: β2-Adrenergic Receptor, NTRKs, Neurokin-1 Receptor, 5-HT Receptors and Dopamine Receptors

Neurotransmitters released from peripheral and autonomic nerves can affect immune cells and targeting neurotransmitter-initiated signaling pathways could become a rational strategy. Neurotransmitters are classified as: 1. amino acids, such as, acetylcholine, glutamate, glycine, gamma-aminobutyric; 2. Biogenic amines, such as, dopamine, norepinephrine (NE), epinephrine and serotonin; 3. Neuropeptides, including, substance P (SP), neuropeptide Y (NPY), and others [128]. Neurotransmitters can also be produced by cancer cells and immune cells. As previously commented, β2-AR on NSCLC inactivates LKB1, increases CREB activity, and enhances tumor-secreted IL-6. ADRB2 mRNA was elevated in NSCLC patients and propranolol (β-AR inhibitor) blocked NE-induced IL-6 [54]. Psychological stress-induced NPY regulates MDSC recruitment to prostate cancers. High levels of NE augment NPY secretion by prostate tumor cells through B2-AR signaling. NPY increases MDSCs in the tumor site, and, induces IL6 release by tumor cells and MDSCs. IL6 activates STAT3 transcription factor. NPY binds to its receptor in tumor cells and in MDSCs [129,130]. MDSC are divided into CD11b^+^Ly6G^low^Ly6C^high^ monocytic MDSCs, and CD11b^+^Ly6G^high^Ly6C^low^ polymorphonuclear MDSCs.

ADRB2 mRNA is upregulated in metastatic prostate cancer [131]. NR-β2-AR signaling further activates focal adhesion kinase (FAK). FAK activation is dependent on the cAMP-PKA signaling pathway. Blocking β2-AR with propranolol, or inhibiting FAK activation, could benefit depressive patients with invasive prostate cancer [132]. Our observations in FaDu and CAL27 HNSCC cell lines are noteworthy, since DHA reduces FAK phosphorylation (Tyr 397) [27]. Another intriguing point is the fact that ADRB2 signaling upregulates nerve growth factor (NGF) and brain-derived neurotrophic factor (BDNF) expression in pancreatic cancer, and increased survival was observed in patients treated with nonselective B-blockers [133]. This opens the gates for screening NGF-BDNF-TRK pathways in cancer patients, and the modulation on IMT is a novel opportunity to augment the effect of anti-PD1 or anti-PD-L1 antibodies. Already, BDNF-TrkB (NTRK2) activation in NSCLC patients has been reported [134]. The NGF receptor TrkA (NTRK1) inhibition decreases YAP-driven transcription [135]. Tumor cells release Yap/Taz-induced growth factors, cytokines, and chemokines, creating an immune suppressive TME that activates and recruits tumor-associated macrophage (TAM), MDSCs, regulatory T cells (Tregs) and cancer associated fibroblasts (CAFs). In addition, Yap/Taz activation translocates PD-L1 onto the plasma membrane where it binds to PD-1 to express on the surface of the cytotoxic T cells (CTLs), and directly inactivates CTLs [136]. YAP is a master transcriptional regulator and, until now, no available drugs have been developed, however, YAP phosphorylation could be abrogated by targeting upstream receptors, such as, NTRK1/NTRK2, or inhibiting YAP phosphorylation (Tyr 357) with repurposing drugs, such as, DHA (Table 2) [27]. Elevated expression of substance P (SP) and its receptor neurokin-1 receptor (NK-1R) has been noted in many cancer types. Importantly, one of the NK-1R antagonists, aprepitant, has been approved by the FDA for the treatment of nausea and emesis caused by chemotherapy. NK-1R mRNA expression is elevated in AML patients. SR140333 (an NK-1R inhibitor) has demonstrated activity in the CML cell line, K562, and inhibits TNF-a, IL-1 and IL-6 expression in K562 cells. ER-mitochondrial calcium overload occurs in response to oxidative stress. The calcium released from the ER is transported to the mitochondria via voltage-dependent anion channel type 1 (VDAC1), which is located in the outer mitochondrial membrane. Both aprepitant and SR 140333 induced ER-mitochondrial calcium overload, with ROS accumulation and cell apoptosis [49].

Serotonin (5-hydroxytryptamine, 5-HT) levels, as well as its receptor, 5HTR_2B,_ are increased in cancer. Platelet derived 5-HT also promotes tumor angiogenesis and tumor growth [137]. Other investigators identified that 5-HTR_2A/C_ are relevant in several pathways and ketanserin is an inhibitor of such activation [138].

qPCR, RNA-Seq or nCounter NanoString based analysis on NNA retrieved from extracellular vesicles from plasma of patients can shed light on the relevance of several biomarkers for prioritizing adjunct treatments that can enhance IMT benefit. We are performing BDNF, NGF, FAK, ADRb2 and NLRP3 assessments by qPCR in extracellular vesicles from plasma and platelets by qPCR in advanced cancer patients and preliminary results show that responders correlate with diminishing levels, mainly, of NGF and BDNF (unpublished data). Dopamine is a precursor for the synthesis of NE. Dopamine binds to five different seven-transmembrane G-protein-coupled receptors, which are divided into two classes: D1-like dopamine receptors (DRs) and D-2 like receptors [137]. Dopamine inhibits NLRP3 (NOD-like receptor family, pyrin domain containing 3) inflammasome activation via D1R (DDR1). DDR1 signaling negatively regulates NLRP3 inflammasome via a second messenger cyclic adenosine monophosphate (cAMP), which binds to NLRP3 and promotes its ubiquitination and degradation via the E3 ubiquitin ligase, MARCH7. This finding suggests that dopamine refrains the development of inflammatory diseases [139].

## 13. Gasdermin D and Gasdermin E

Innate immune cells sense pathogenic infection or tissue injury by PAMPs and Damps, respectively, through a range of PRRs. Macrophages survey for potential infection or cellular injury via PRRs, including membrane bound TLRs or cytososlic NOD-like receptors (NLRs). Recognition of PAMPs by TLRs induces signaling transduction of the NF-kB pathway with cytokine expression. Activation of cytosolic PRRs, such as NLRP3, leads to the formation of a multiple protein complex (inflammasome). Extracellular ATP is a DAMP that activates NLRP3 by triggering K+ efflux via binding to purinergic P2X7 receptor (P2X7R). ATP-PX7R-induced K^+^ efflux can be mediated by TWIK2 (two-pore domain weak inwardly rectifying K^+^ channel 2). The triggering of NLRP3 recruits the adaptor ASC (apoptosis-associated speck-like protein containing a caspase recruitment domain) and pro-caspase-1 to form a large multi-protein NLRP3 inflammasome, resulting in the autocatalytic activation of caspase-1. Activated caspase-1 cleaves gasdermin D (GSDMD) into C-terminal and N-terminal (GSDMN-NT) fragments. GSDMD-NT binds to and forms pores on the plasma membrane, causing pyroptosis-a rapid proinflammatory programmed cell death with cellular swelling, membrane disruption, and release of proinflammatory contents, including, HMGB1 [140] (Figure 3).

Upon apoptotic stimulation, cleavage of gasdermin E (GSDME/DFNA5) by caspase-3 produces GSDME-NT (−37kDa) forming pores on the plasma membrane, and, thereby, transforms apoptosis into pyroptosis/necrosis [140] (Figure 3). In RAW 264.7 cells deficient in ASC, ATP induced a delayed form of pyroptotic cell death that relies, in part, on GSDME cleavage. In BMDM, ATP induces rapid pyroptosis mediated by GDSMD cleavage, whereas, it induced delayed pyroptosis associated with GDSME cleavage when the canonical NLRP3 pathway is blocked [140] (Figure 3).

There are nine members of the IL-1 family, the best known are IL-1a, IL-1β and IL-18. IL-1 proteins do not contain N-terminal secretion signal sequences and are not released into the extracellular space through the conventional secretory pathway. The pore-forming protein GSDMD induces the release of IL-1 from living (hyperactive) or dead (pyroptotic) cells. At the end of a short LPS triggered pathway (LPS sensing activates caspase 4 and caspase-5 to cleave GSDMD, causing membrane permeabilization and subsequent NLRP3 inflammasome assembly. The NLRP3 inflammasome stimulates caspase 1 to unleash a second wave of GSDMD activity and the sole wave of pro-IL-1 processing [141]. Disulfiram (Antabuse) has been identified as a GSDMD inhibitor representing an attractive therapeutic approach [142], as well as, the NLRP3 inhibitor, OLT1177 (Dapansutrile) (Table 2) [50,143]. High expression of NLRP3 has been seen in melanoma patients resistant to pembrolizumab. OLT1177 is a safe, orally active treatment for acute gout flares. A pembrolizumab plus OLT1177 study is ongoing. Anakinra inhibits binding of both the IL-1α and IL-1β to IL-1 receptors. IL-1-associated inflammatory signature has been reported in breast cancer.

Conversely, GSDME suppresses tumor growth by activating tumor-infiltrating NK and CD8^+^ T killer lymphocytes. Noteworthy is the fact that the GSDME-positive tumors do not release IL-1β. It is noted that cancer cells develop epigenetic suppression of GSDME expression and LOF mutations to avoid GSDME-mediated tumor suppression. Cancers with GDME mutations reduce pyroptosis, and mutations of D270, the shared granzyme B/caspase 3 cleavage site, enable tumors to evade tumor suppression by GSDME. Decitabine, a DNA methylation inhibitor, can induce GSDME [51]. The findings of Lieberman [51] and those from Dinarello are crucial for clinical implementation. GSDMD and GSDME can serve as important biomarkers in predicting response to anti-PD-1 or anti-PD-L1 antibodies, and several IL-1 or NLRP3 repurposing compounds could be critical for the IMT efficacy.

## 14. K-Ras Mutations

Upon stress, the solute carrier family 7, member 11 (SLC7A11) promoter is activated by NRF2 and ATF4 (Figure 4). Both proteins are altered in K-Ras mutant NSCLC [111]. Overexpression of the cancer stem cell marker CD44 increases the stability of SLC7A11 by promoting the interaction between SLC7A11 and OTUB1 [144]. OTUB1 was reported to interfere with DNA repair by suppressing UBC13, therefore, limiting RNF168 function [145]. OTUB1 can also stabilize FOXM1. Henceforth, the SLC7A11/CD44 and CD44 complex can abrogate ferroptosis and presumably have an important role in NSCLC with K-Ras mutations. Soluble CD44 (sCD44) secreted by breast cancer cells induces IL-1β secretion by macrophages, highlighting that the sCD44-1L-1β axis could be considered in IMT [146].

One of the key findings to understand the limited efficacy of IMT is that the activation of β-catenin signaling results in T-cell exclusion. WNT/β-catenin pathway signaling can contribute to 48% of melanomas. In tumors without active B-catenin signaling, ATF3 is not induced, and CCL4 is transcribed and secreted. Downstream CD103^+^ dendritic cells are attracted with subsequent activation of CD8^+^ T cells. However, in tumors with active β-catenin signaling, ATF3 suppresses CCL4 transcription with immune escape through a lack of dendritic cells [147]. Recently, adenosine A1 receptor (ADORA1) has been reported to evade immune response by regulating the ATF3-PD-L1 axis [148]. Higher ADORA1, lower ATF3, and lower PD-L1 expression levels were noted in tumor tissues from non-responders among anti-PD-1 antibody (nivolumab)-treated NSCLC patients [148]. Extracellular adenosine functions as a ligand that binds to adenosine receptors, such as the ADORA2A, promoting tumor growth by recruitment of MDSCs and ADORA1 that signals downstream of cAMP/PKA/CREB/ATF3/PD-L1 (Figure 4). ADORA1 antagonists could enhance the effect of IMT [148]. ADORA1-ATF3-PD-L1 merits further scrutiny in K-Ras mutant NSCLC cell lines and patients. Chemotherapy in TNBC with, either, carboplatin, paclitaxel or gemcitabine, increases extracellular adenosine that is catalyzed by the 5’-ectonucleotidase CD73. Transcripts of CD73, CD47, PD-L1, HIF-1a are increased following treatment with each of the abovementioned cytotoxic drugs. In the work of Semenza’s group, A2A receptor signaling reduces the activity of CD8^+^T cells (with a decrease in IFN-g mRNA levels) and NKs, it also increases T_regs_ and MDSCs [149]. HIF inhibitors could block the counter therapeutic effect of paclitaxel and other cytotoxic chemotherapy agents [149].

The effect of cytotoxic drugs is complex. For instance, paclitaxel, due to induction of mitotic arrest, allows cGAS to activate IRF3 [150]. Intriguingly, gemcitabine could be of benefit in K-Ras mutant NSCLC with LKB1^nsm^ or deletion. Loss of expression of LKB1 enhances the pool of cytidine deaminase, a key enzyme that metabolizes gemcitabine following its uptake [151].

## 15. Transforming Growth Factor-β Signaling in K-Ras Mutations

K-Ras mutant NSCLC with LKB1^nsm^ or LKB1 loss has poor prognosis, as above commented. The loss of LKB1 in NSCLC leads to the dephosphorylation of CRT2 that displaces to the nucleus where it binds to CREB over the ID1 promoter and stimulates ID1 expression (Figure 4) [29]. Repulsive guidance molecules (RGMs) are co-receptors of bone morphogenetic proteins (BMPs) and programmed death ligand 2 (PD-L2). Low RGMB, observed in NSCLC, correlates with high expression of Snail, FAK and ID1 [152]. K-Ras mutant pancreatic cancers with intact transforming growth factor-β (TGF-β) pathway drive tumor growth via ID1. K-Ras pancreatic cancers with active TGF-β pathways (50%) upregulate the Snail family transcriptional repressor 1 (SNAIL), thus repressing KLF5 expression and, in parallel, activating AKT switch ID1 regulation by TGF-β from repression into induction. ID1 inhibits the activation of E12/E47. It is predicted that ID1 dependent pancreatic cancers could be sensitive to specific ID1 small-molecule inhibitors [153]. The potential benefit of gefitinib in suppressing ID1 in K-Ras mutant NSCLC also warrants investigation. As commented earlier, ID1 over-expression induces necroptosis during gefitinib treatment through RIP3/MLKL upregulation [28].

In TGF-β treated NSCLC cells (A549, H460, H1299), active PLK1 phosphorylated at T210 is abundant and upregulates many genes related to TGF-β signaling. Loss of PLK by specific inhibitors (volasertib and poloxin) blocks tumor growth and metastatic activity induced by TGF-β. In TGF-β treated A549 cells, the mRNA levels of TNFAIP6 were increased. TNFα-stimulated gene 6 (TSG6) encoded by TNFAIP6 is involved in EMT through effects on hyaluronan structure and CD44-dependent triggering of cell responses. High expression of both TNFAIP6 and PLK1 could be considered therapeutic targets in NSCLC, including K-Ras mutant, as inferred from the study [154]. In colorectal cancer, the combined use of a TGF-β-activated kinase 1 (TAK1) inhibitor plus a TGFBR1 inhibitor avoid the IL-1β and TGF-β1-mediated conversion of resident fibroblasts into cancer-associated fibroblasts (CAFs) [155]. Celastrol used for the treatment of asthma, rheumatoid arthritis, and neurodegenerative disease, inhibits TAK1 activation. Celastrol reduced TNF-α-induced and TGF-β1-induced NF-кB activation. The inhibition is mediated by reducing TAK1 phosphorylation and IKK activation in HNSCC cells [156]. Celastrol could be a cheap, safe drug targeting the TAK1-NF-кB pathway, and others, where TAK1 is regulating [157]. MED12 and BAMBI negatively regulate TGF-β receptor. Loss of MED12 or BAMBI induces an EMT-like phenotype in NSCLC, enhancing immunosuppressive effects [158,159,160]. Epigallocatechin gallate (EGCG), a food supplement, is a fibroblast-specific, irreversible inhibitor of both lysyl oxidase-like 2 (LOXL2) and TGF-β receptors 1 and 2 (TGBR1/2) [161]. Inhibition of LOXL2 generates the TGFBR1/2 kinase inhibitor [162] and lowers cellular levels of phosphorylated SMAD2/3. Twenty patients with pulmonary fibrosis were treated with EGCG (Teavigo) capsules at a daily dose of 600 mg for 14 days before undergoing lung biopsy. In the treated patients, tissue levels of SNAIL and phosphorylated SMAD3 were lower than those in the control group [161]. Bone-marrow-derived cells (BMDCs)-derived extracellular vesicles (EVs) increase the formation of liver pre-metastatic niche in lung cancer. Secretion of EV miR-92 by BMDCs suppresses SMAD7, leading to the enhancement of TGF-B signaling in hepatic stellate cells [163]. A TGF-β signature predicts clinical outcomes to durvalumab in HIV-1 cancer patients [164]

## 16. Metabolism-Based Therapy for LUAD with K-Ras and LKB1 Mutations

As previously commented, the overall survival of patients with LKB1^nsm^ is extremely poor, as is confirmed in a large group of patients from Cologne, with median overall survival of 3.2 months versus 10 months for those with wild-type LKB1 [165]. Therefore, solutions are urgently needed. Metformin could serve as a potential adjunct therapy, as has been noted in EGFR mutant NSCLC patients receiving gefitinib plus metformin, versus gefitinib alone, in experimental gastric cancer models [34,166]. The ID1 expression levels can also help for potential intervention with EGFR TKIs. Several years ago, it was underlined that LKB1 inactivation could respond to the diabetes biguanide compounds, metformin and phenformin, that inhibit complex I of the mitochondria. Selective response to phenformin was observed in K-Ras mouse models of NSCLC with LKB1 mutations, but not those with K-Ras and p53 mutations [167]. A synergistic effect was detected in a phase II trial of sorafenib (multi-kinase inhibitor) in K-Ras mutant NSCLC patients that were also receiving metformin. AMPK activation is achieved with metformin, but also with salicylates. Sorafenib induces ER stress, Ros and calcium release, leading to the expression of calcium activated kinase (CAMKK2) that activates AMPK at threonine 172 (Thr 172) [168]. The combination of sorafenib and acetylsalicylic acid significantly reduced cell proliferation over 72 h in the K-Ras mutant cell line, A549. The combination of sorafenib and aspirin increases activation of ERK1/2 and the phosphorylation of the AMPK substrate acetyl-coenzyme A carboxylase (ACC). The combination was strongly suggested to be useful in the treatment of K-Ras mutant NSCLC [169]

K-Ras/LKB1^nsm^ cells express the urea cycle enzyme carbamoyl phosphate synthetase-1 (CSP1), which produces carbamoyl phosphate in the mitochondria from ammonia and bicarbonate, initiating nitrogen disposal. Transcription of CPS1 is suppressed by LKB1 through AMPK and CSP1 mRNA and protein expression correlates inversely in NSCLC (Figure 4) [170]. This primal study illustrates that metabolic alterations mediated by concurrent mutations of K-Ras and LKB1 render cells dependent on CPS1 for pyrimidine synthesis. It is supposed that the dependence on CPS1 is exacerbated by mutant K-Ras due to its action on glutamine metabolism. These findings identify CSP1 expression as a sensor for gauging the evolution of K-Ras with LKB1^nsm^ or LKB1 deficiency and, as above explained, intervening with the glutamine pathway (for example with DHA) or restoring LKB1 with metformin, could be further investigated.

Metformin (N,N-dimethylbiguanide) activity relies also on inhibition of fructose-1-6- bisphosphatase-1 (FBP1) by regulating hepatic glucose production, reducing HGF and normalizing blood glucose levels in hyperglycemic type 2 diabetics [171]. Plasma metformin concentration in persons is markedly variable, in part, due to differences in intestinal metformin transporter organication transporter 1 (OCT1). Furthermore, severe hyperglycemia is manifested in hepatic LKB1-null mice due to impaired activity of the AMPK-related kinase, SIK. In the LKB1-and SIK proficient state, LKB1 promotes the activity of SIL1/3, which alters the activity of SIK1/3, altering SIK dependent genes and reducing lung tumor growth. In the absence of LKB1, SIK1/3 activity is reduced, but promotes tumor growth (Figure 4). Finally, when SIK activity is completely lost, tumor growth increases even more than upon LKB1 loss [172]. Another study found that AREG (an EGFR ligand) was upregulated in LKB1-deficient tumors in a SIK1/3-CRTC2 dependent manner [173].

In LKB1-deficient lung cancer, glutamate dehydrogenase 1 (GDH1) is upregulated by pleomorphic adenoma gene 1 (PLAG1). The GDH1 product alpha-ketoglutarate activates calcium/calmodulin-dependent protein kinase kinase 2 (CamKK2) and activates AMPK contributing to anoikis resistance and tumor metastasis (Figure 4). GDH1 knockdown or treatment with selective GDH1 inhibitor (R162) sensitized LKB1-deficient A549, H157 and H460 cells to anoikis induction [174]. Metformin activity is due to multiple mechanisms and warrants in depth investigation for clinical applicability. Biguanides, phenformin and metformin, inhibit growth by inhibiting mitochondrial respiratory function with limits the transit of the RAgA-RagC GTPase heterodimer through the nuclear pore complex. Nuclear exclusion renders RagC unable to stimulate mechanistic target of rapamycin complex 1 (mTORC1). Metformin-induced inactivation of mTORC1 subsequently inhibits growth through transcription of acyl-CoA dehydrogenase family member-10 (ACAD10). Both limited nuclear pore transit and upregulation of ACAD10 are required for biguanides to reduce viability in melanoma and pancreatic cancer cells and to extend C. elegans lifespan [175].

SYMD3, a member of the SET and MYND-domain (SMYD) family, is involved in gene transcription regulation and tumor progression as histone lysine methyltransferase. In cancer cell lines, SMYD3-mediated methylation of MAP3K2 at lysine 260 enhances activation of the Ras/Raf/MEK/ERK signaling and SMYD3 depletion synergizes with a MEK inhibitor to block Ras-driven tumors. SMYD3 is often overexpressed in pancreatic and lung cancer [176]. Recently, it has been seen that SMYD3 binds to CDK2 and MMP2 promoter, increasing gene transcription I hepatocellular carcinoma. Pharmacologically targeting SMYD3 with BCI-121 inhibitor represses hepatocellular carcinoma cells [177].

## 17. BACH1 and NRF2-KEAP Alterations in K-Ras Driven LUAD

BTB and CNC homology 1 (BACH1), a heme-binding transcription factor, targets the mitochondrial metabolism. BACH1 mRNA expression is increased in several classes of cancer, including lung cancer. BACH1 negatively regulates transcription of electron transport chain (ETC) genes and can hamper the activity of metformin (ETC inhibitor) (Figure 4). Combination treatment with hemin (active ingredient of panhematin, a FDA approved drug to treat acute porphyria) and metformin suppresses tumor growth in BACH1 expressing TNBC cell lines, or patient-derived xenografts. Hemin is given for ten days to degrade BACH1 before metformin [178]. Simultaneously, BACH1 promotes progression and metastasis in LUADs. NRF2 upregulation, due to the loss of KEAP function, accumulates BACH1 via induction of heme oxygenase. It is proposed that drugs targeting the heme pathway could be a therapeutic option in LUAD patients with dysregulation of KEAP-NRF2 pathway, often mutated in K-Ras mutant NSCLC [179]. In addition, fructosamine-3-kinase (FN3K) regulates NRF2 function (Figure 4). FN3K promotes hepatocellular cancer by mediating deglycation of NRF2. De-glycation, removal of attached sugars, is triggered by FN3K and is essential in NRF2-driven cancers [180]. L-Butathione sulfoximine (BSO) inhibits the NRF2 target gene, γ-glutamylcysteine synthetase (GSC) [40]. Hence, drugs targeting the glutathione redox system (see ferroptosis) should be taken into account for the treatment of K-Ras mutant LUADs with NRF2, KEAP, BACH1 and FN3K, which are important predictive factors for ideal therapy (Figure 4, Table 2).

NRF2-KEAP alterations in K-Ras driven LUADs are also dependent on increased glutaminolysis, and offer another therapeutic opportunity through the inhibition of glutaminase [181]. CB-839 has been used in lung squamous cell cancer [182] and glutamine blockade can overcome tumor immune evasion [183]. Recently, Heymach’s group further stressed that, in K-Ras/LKB1 LUADS, the KEAP-NRF2 pathway confers an enhanced sensitivity to the glutaminase inhibitor CB-839 in vitro and in vivo [184]. As previously explained, LKB1 mutations are refractory to anti-PD-1 antibodies [30]. The effect of the MEK inhibitor (selumetinib) has also been discouraging [185].

Mutations in NRF2 and KEAP1 were reported for the first time in 12 cell lines and 54 NSCLC samples. KEAP1 mutations were seen in 50% of cell lines and 19% of tumor specimens [186]. KEAP mutations are homozygous in A549, H460, H838, and H1435, and heterozygous in H1395 and H1993. In the A549, H838 and H1435, G-T transversions were probably induced by tobacco smoke. KEAP mutations showed increased NRF2 staining in the nucleus and cytoplasm [186]. Of interest is the fact that partner and localizer of BRCA2 (PALB2) interacts with KEAP1, repressing its function and promoting NRF2 nuclear accumulation and function (Figure 4) [187]. We observed that high levels of PALB2 mRNA expression conferred sensitivity to docetaxel in advanced NSCLC patients. PALB2 was predictive of PFS, overall survival, and response among 158 patients. Patients with high PALB2 mRNA have a response rate of 77% to cisplatin-docetaxel, compared with a response rate of 23% for those with low PALB2 mRNA expression (*p* = 0.04) [188]. Knock-down of PALB2 in EBC1 and H1993 cells diminish the antiproliferative effect of docetaxel [188]. Therefore, docetaxel could be a therapeutic tool in the treatment of K-Ras mutant NSCLC with KEAP mutations, and PALB2 mRNA expression could serve as a biomarker. NRF2 regulates proteins that support NRF2 transcriptional output, such as NR0B1, and they have been identified as drugabble [189]. Another, potentially therapeutic target is, ASF1, a histone H3-H4 chaperone involved in gene regulation and DNA replication. Tumor cell-intrinsic Asf1a deficiency promotes inflammation and M1-like macrophages via secretion of GM-CSF, with T-cell activation. ASF1A loss, combined with anti-PD-1 antibodies, exerts a significant tumor growth inhibition in K-Ras LUAD models [190].

## 18. Perspectives: Potential Role for PDCD1 and PDCD1LG1 Genes, MICA and MICB, Exosomes and Other Checkpoints

Since the advent of IMT [2], translational research has gravitated around IMT, nevertheless, clinical outcomes are still meager [191] and the recommendations for treatment of NSCLC relies on anti-PD-1 antibodies (pembrolizumab) or anti-PD-L1 antibodies (atezolizumab) with or without chemotherapy [5]. PD-L1 IHC testing is still considered a predictive biomarker according to the IASLC Pathology Committee [192]. However, it is described that tumor cells express PD-1 and PD-L1, which inhibit tumor cell growth by impairing AKT and ERK1/2 pathways and preventing the interaction with PD-1-expressing T cells [191]. Tumor cells expressing PD-1/PD-L1 are resistant to antibodies targeting PD-1/PD-L1 therapy [193]. The study of Wang and Yang [193] described that the PDCC1 gene encoding PD-1 is transcribed by cancer cells. PD1 is expressed in NSCLC cell lines around 55 KDa in size, which is similar to the size of T-cell-expressed PD-1. Flow cytometry shows that PD-1 is expressed in a subpopulation of cancer cells. Simultaneous overexpression of PDCD1 and PDCD1LG1 in a subpopulation of cancer cells can inhibit cell proliferation and signaling transduction. Calu-1, SW480, HT-29, BxPC-3, SK-BR-23 and U-2 OS cells when treated with nivolumab, pembrolizumab or atezolizumab have increased proliferation with higher levels of AKT and ERK phosphorylation. Until now it was recognized that PD-1 is mainly expressed on the activated T cells, B cells, and monocytes. Several studies alerted that PD-1 is expressed on tumor cells as well as based on the transcriptomic and proteomic data. Moreover, murine tumor cells expressing PD-1 exhibit increased growth under PD-1 targeted antibody treatment both in vitro and in vivo, suggesting that tumor cell-intrinsic PD-1 plays an antitumor role in NSCLC [194]. A NSCLC patient with evident cancer cell-intrinsic PD-1 expression progresses rapidly after pembrolizumab treatment [194]. In sum, the model of Wang and Yang [193] identifies tumor-cell intrinsic PD-1/PD-L1 antitumor function in the absence of adaptive immunity. When antibodies activate T cells, tumor cells are destroyed by the activated T cells, however hyper-progression occurs after anti-PD-1 or anti-PD-L1 antibodies activate tumor cells and overwhelm activation of T cells [193].

Ceaseless translational research is warranted in order to improve the outlook of cancer patients. Natural killer (NK cells), CD8^+^ T cells and gamma delta T cells express NKG2D. NKG2D ligands (the major histocompatibility complex (MHC) class I-related molecules A and B (MICA and MICB) are of interest. The lack of NKG2D ligands (MICA, MICB, or others) is a trait of resistant leukemia stem cells. Poly-ADP-ribose polymerase 1 (PARP1) represses the expression of NKG2D ligands and targeting resistant leukemia stem cells with PARP inhibitors could possibly restore the NK function [195]. Warfarin (an anticoagulant) interferes withGas 6, the ligand of TAM tyrosine kinase receptors, Tyro, AXL and Mer (TAM). TAM activates the E3 ubiquitin ligase CBl-b (casitas B-lineage lymphoma-b), blocking NKG2D receptors. Therefore, warfarin or TAM inhibitors can activate NK cells [196]. BCL11B is a zinc finger transcription factor that upregulates MICA/MICB expression by acting as a competitive endogenous RNA [197]. The bacterial abundance and composition in the bronchoalveolar lavage fluid showed that the lungs from tumor-bearing KP (K-ras mutation and p53 loss) mice harbored unique commensal communities and depletion of commensal microbiota suppresses LUAD development [198]. Exosomes, or extracellular vesicles, are an important piece of the puzzle in cancer progression and immune evasion. In vitro, the function of dendritic cells is repressed by EGFR exon 19 deletion in Lewis lung cancer. Through exosome uptake, exosomes derived from the EGFR-19 deletion Lewis lung cancer cells could transfer EGFR-19 deletion to the surface of dendritic cells. The combination of gefitinib and GM-CSF treatment recover T cell infiltration in EGFR exon 19 deletion tumors [199]. The assessment of EGFR mutations, for example, in plasma exosomes in NSCLC patients could further reveal the relevance of the experimental data. Tumor derived extracellular vesicles downregulate type I interferon receptor and cholesterol 25-hydroxylase (CH25H). It is noteworthy that reserpine (an antihypertensive drug) inhibits extracellular vesicle uptake and metastasis [200].

Besides PD-L1, PD-1 can bind to another B7 family member, PD-L2, that is expressed on DCs, macrophages, and, at less frequency than PD-L1, on cancer cells, including NSCLC. PD-1-PD-L2 interaction has been proposed to primarily inhibit the CD4 + T helper 2 subsets (Th2) response, whereas PD-L2 interaction with RGMB receptor results in co-stimulation of T CD4 + cell responses, and promotes T helper 1 subset (Th1) polarization. Similar to PD-L1, its expression can be induced by immune cell-secreted inflammatory cytokines, such as, IFNs, IL-4, IL-10, and granulocyte/macrophage colony-stimulating factor, but also by intracellular oncogenic signaling that can contribute to intrinsic immune resistance. PD-L2 expression significantly correlated with PD-L1 expression in different NSCLC gene expression datasets, and both PD-L1 and PD-L2 expression were associated with specific gene signatures, suggesting a potential role for PD-L2 and these gene expression biomarkers in predicting clinical responses to immune checkpoint blockade [201].

Beyond PD-1, additional immune checkpoint molecules, such as, lymphocyte-activation gene-3 (LAG-3) and T-cell immunoglobulin and mucin domain-3 (TIM-3), can be activated after T-cell receptor stimulation, and mediate T-cell suppression. These checkpoint proteins can be co-expressed on cancer-specific immune cells, although it has been demonstrated that they may function through nonoverlapping molecular mechanisms to regulate anti-tumor immune response. Targeting multiple complementary immune pathways may activate the antitumor immune response more effectively and improve clinical benefit. Clinical studies with TIM-3 or LAG-3 inhibitors, alone, or in combination with other drugs, including other immune checkpoint inhibitors, are ongoing. Recent data suggest a potential role of these checkpoints in predicting clinical benefit from anti-PD1/PD-L1 therapy. Indeed, an elevated expression of LAG-3 on T cells in baseline samples of NSCLC patients treated with PD-1 axis inhibitors was significantly associated with shorter PFS [202]. The association of LAG-3-and PD-1-inhibitors may be able to overcome resistance and enhance responses, as suggested by preliminary results of the CA224-020 clinical trial (NCT01968109) including melanoma patients progressed on prior anti-PD-1/PD-L1 therapy. In this study, LAG-3 expression on immune cells was predictive of clinical benefit from this combination [203]. Phase 2 studies with the anti-LAG-3 inhibitor, relatlimab, in association with nivolumab are ongoing in early stage and advanced-NSCLC (NCT04205552, NCT02750514).

Finally, other activated oncogenes may play an important role as predictors of response to immune checkpoint blockade therapy. Recent data suggest immunotherapy has favorable activity in patients with both BRAF V600E and non-V600E mutations. BRAF mutations were associated with high level of PD-L1 expression (42 to 50% of cases), low/intermediate TMB and microsatellite-stable status [204]. A limited number of patients included in the study had co-expressing tumors (BRAF V600E mutations and PD-L1 > 50% of cancer cells). For these patients, the use of targeted therapy was associated with better clinical outcome than immunotherapy [204]. Alterations in ARID1A gene, encoding for AT-Rich Interactive Domain-containing protein 1A involved in chromatin remodeling, have been shown to impair mismatch repair (MMR), which may in turn cause an increased mutation burden and drive responses to immune checkpoint inhibitors [205]. In a recent study, ARID1A alterations assessed by tissue next-generation sequencing in different cancer types, including NSCLC, were significantly associated with longer PFS after checkpoint blockade, regardless of microsatellite instability or mutational burden, suggesting that the role of this biomarker need to be further explored in clinical setting [206].

## 19. Conclusions

Deciphering the molecular composition of NSCLC requires the understanding of multiple biological cues that can enhance the benefit of treatment, in general, and also in the different molecular subclasses of cancer. Each entity, either with druggable driver mutations, or not, requires a more fundamental anti-cancer treatment and a profound re-formulation on the present view of treatment. Tumor PDCC1mRNA and PD-1 proteins surface as critical parameters and, ideally, IMT should be prescribed via tailored therapy.

## Figures and Tables

**Figure 1 cancers-12-01475-f001:**
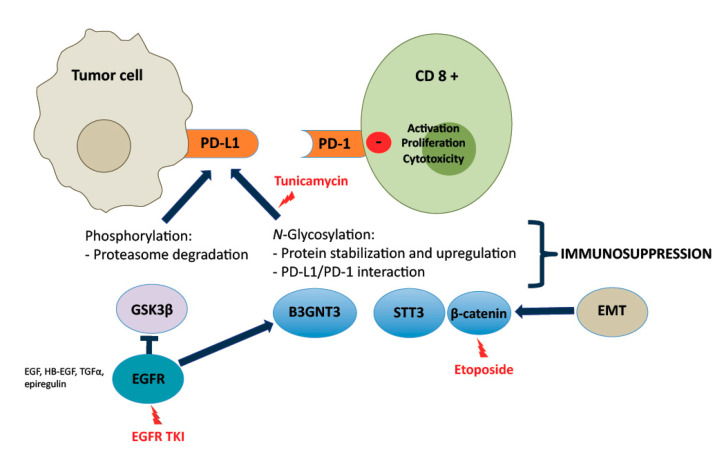
PD-L1 glycosylation and selected proteins involved in the mechanism of glycosylation and phosphorylation of PD-L1. Glycogen synthase kinase 3β (GSK3β), phosphorylates PD-L1, leading to its ubiquitination. EGFR, activated by its ligand EGF, epiregulin, TGFα, and heparin-binding EGF, induces PD-L1 expression by inhibiting GSK3β. EGFR tyrosine kinase inhibitors (TKIs) can prevent this mechanism. β-1,3-N-acetylglucosaminyl transferase (B3GNT3) is involved in PD-L1 glycosylation, and it is upregulated by EGFR. N-glycosyltransferase STT3 positively regulates PD-L1 glycosylation; Epithelial–mesenchymal transition (EMT) induces N-glycosyltransferase STT3 through β-catenin. N-linked glycosylation can be inhibited by tunicamycin. Etoposide, reduces nuclear β-catenin and inhibits the EMT-β-catenin-STT3-PD-L1 signaling.

**Figure 2 cancers-12-01475-f002:**
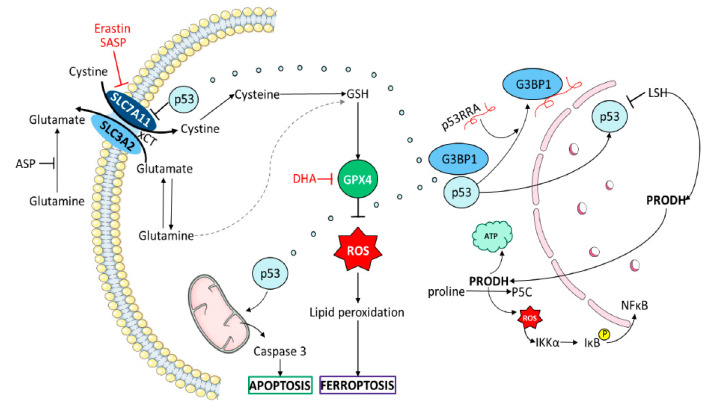
Ferroptosis is the result of an accumulation of lipid reactive species. ROS production is inhibited by metabolites derived from the amino acid cysteine, and glutamine by the antiporter xCT (−) transporter/SLC7A11. In addition, the p53-related lncRNA (P53RRA) is also involved in the regulation of SLC7A11. For a full explanation, see the section on Glutathione synthesis and ferroptosis cystine-glutamate antiporter (xCT)-GSH/GPX pathway and p53.

**Figure 3 cancers-12-01475-f003:**
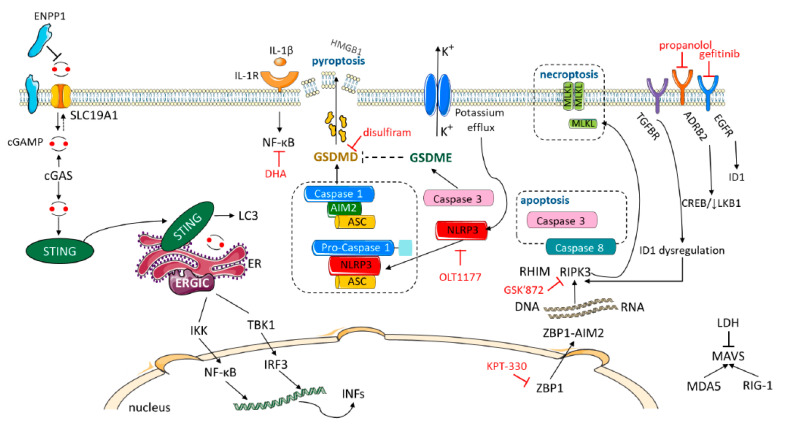
Necroptosis, pyroptosis and cGAS signaling pathways. Necroptosis is a form of programmed necrosis with sequential activation of the receptor-interacting protein kinase 1 (RIPK1), RIPK3 and mixed-lineage kinase-domain-like (MLKL) protein. MLKL is a pore-forming protein that promotes necroptotic death, lytic inflammatory death, by releasing intracellular molecules such as HMGB1 and IL-1alpha. RIPK1 deficiency or mutations of its RIP homotypic interaction motif (RHIM) triggers Z-DNA-binding protein (ZBP1)-dependent necroptosis and inflammation. Treatment of cells with a nuclear export inhibitor (KPT330) (in red) or a RIPK3 inhibitor (GSK’843) (in red) avoids MLK-dependent necroptosis. Cell death is also negatively regulated by caspase 8. Cleavage of RIPK1 by caspase 8 is essential for neutralizing RIPK3 and MLKL. Following MLKL-mediated nuclear rupture, HMGB1 can also rapidly exit the nuclei of cells undergoing TNF-alpha-stimulated necroptosis. Dysregulation of the DNA binding 1 (ID1) inhibitor by alterations in either EGFR, TGF-β or ADRB2 signaling, as well as loss of LKB1, can also activate RIPK3. Propranolol (ADRB2 inhibitor or gefitinib (EGFR TKI) could correct the ID1-induced RIPK3 activation. MLKL causes potassium efflux that activates the nucleotide-binding oligomerization domain (NOD)-like receptor protein 3 (NLRP3) inflammasome to recruit the adaptor protein apoptosis-associated speck-like protein containing caspase activation and recruitment domain (ASC) and triggers caspase-1 processing of IL-1β. Activation of caspase-1 also cleaves gasdermin D, gasdermin N-terminal binds to and forms pores on the plasma membrane causing pyroptosis, a form of cell death that releases proinflammatory contents, including HMGB1. Gasdermin E acts as a tumor suppressor. Disulfiram (red color) could act as a gasdermin D inhibitor and OLT1177 (red color) as an NLRP3 blocker. Cytoplasmic DNA can also activate absent in melanoma 2 (AIM2)-ZBP1 inflammasome signaling. Cytosolic DNA stimulates cyclic GMP-AMP synthase (cGAS)-cyclic GMP-AMP (cGAMP)-stimulator of interferon genes (STING). STING dimerizes in the endoplasmic reticulum (ER) and translocates to the ER-Golgi intermediate compartment (ERGIC), leading to TBK1-IRF3-type I interferon signaling. The ectonucleotide pyrophosphatase phosphodiesterase (ENPP1) blocks the export of cGAMP to the extracellular space. Inhibitors of ENNP1 can boost extracellular cGAMP concentration and enhance the immune system. SLC19A1 acts as a cGAMP importer and thereby activates STING. Lactate (LDH) inhibits mitochondrial antiviral signaling (MAVS). Other abbreviations and symbols: cGAMP (symbol as two dots with linkers). Retinoic-acid-inducible gene I (RIG-I).

**Figure 4 cancers-12-01475-f004:**
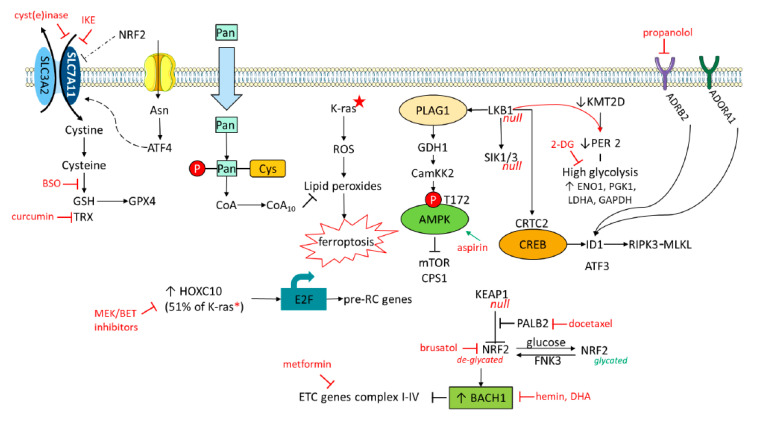
K-ras transcription regulation roadmap. Mutant K-ras signaling induces increased production of reactive oxygen species (ROS) that is counterbalanced by thioredoxin (TRX) and glutathione (GSH) synthesis. GSH leads to the formation of the lipid peroxide-detoxifying enzyme glutathione peroxidase 4 (GPX4). Targeting cystine input with an antiporter xCT (SLC7A11) inhibitor, imidazole ketone erastin (IKE), or cotreatment with an inhibitor of GSH synthesis, buthionine sulfoximine (BSO), induces lipid oxidation reducing cell viability. Coenzyme A (CoA) is produced also by cysteine via the pantothenate (Pant) pathway. CoA and its derivative coenzyme Q10 (CoQ10) cooperate by inhibiting accumulation of lipid ROS (ferroptosis). In vitro cyst(e)inase treatment reduces cell viability in multiple tumor models. Co-mutations and/or loss of LKB1 and loss of salt-inducible kinase 1 and 3 (SIK1/3) activate CRTC2 that, with CREB, could signal via ID1 to RIPK3-MLKL (see also Figure 3). CREB is activatable by adenosine A1 receptor (ADORA1) and ADRB2. It is possible that loss of LKB1 with inactivation of AMP-activated protein kinase (AMPK) can downregulate Period (Per), a circadian gene, that is also down regulated when the histone methyltransferase KMT2D is diminished (by mutations occurring often in lung cancer. Loss of KMT2D and Per lead to increased expression of numerous glycolytic genes (ENO1, PGK1, LDHA1, GAPDH and others). Co-treatment up-regulation of genes is involved in the glycolysis pathway. Also, when LKB1 is impaired, pleomorphic adenoma gene 1 (PLAG1) increases glutamate dehydrogenase 1 (GDH1), which leads to increased calcium calmodulin-dependent protein kinase kinase (CamKK2) with AMPK T172 phosphorylation impinging on mTOR and CPS1 (see the text). Aspirin can restore AMPK function. KEAP mutations or loss of function are closely related to NRF2 overproduction. PALB2 is a negative regulator of KEAP function. Docetaxel was associated with better progression free survival, overall survival and response rate in metastatic NSCLC patients with high PALB2 mRNA (see text). De-glycation of NRF2 by fructosamine-3-kinase (FN3K) is pivotal for NRF2 function. Brusatol is a NRF2 inhibitor. BTC and CNC homology 1 (BACH1) is induced by alterations in the KEAP- NRF2 axis. BACH1, a heme-binding transcription factor, negatively regulates transcription of electron transport chain (ETC) genes. Combination of hemin (we posit that also dihidroartemisin (DHA)) with metformin is effective in cancer cell lines. The homeobox protein HOXC10 is overexpressed in 51% of K-ras mutant NSCLC); BET plus MEK inhibitors suppress pre-replication, triggering stalled replication and DNA damage. Proviso: The figure construction is based on robust findings, some of them recently described, and thereby cited in the legend. Since there are multiple biological routes involved, the pretension of the figure is to provide a glimpse at the approximate actual knowledge that could serve for developing biomarkers and customizing therapy in K-ras mutant patients.

**Table 1 cancers-12-01475-t001:** Anti-PD1/PD-L1 antibodies approved for clnical use.

Anti-PD1/PDL1 Antibody	FDA-Approved Indications
Pembrolizumab	Melanoma, NSCLC, SCLC, HNSCC, cHL, PMBCL, urothelial carcinoma, MSI-H or dMMR cancer, gastric cancer, esophageal cancer, cervical cancer, endometrial carcinoma, RCC, hepatocellular carcinoma and Merkel cell carcinoma
Nivolumab	Melanoma, NSCLC, SCLC, RCC, cHL, HNSCC, urothelial carcinoma, MSI-H or dMMR colorectal cancer and hepatocellular carcinoma.
Atezolizumab	Urothelial carcinoma, NSCLC, TNBC, SCLC
Durvalumab	Urothelial Carcinoma, NSCLC, SCLC
Avelumab	Merkel cell carcinoma, urothelial carcinoma, RCC
Cemiplimab	Cutaneous squamous cell carcinoma

Abbreviations: NSCLC: non-small cell lung cancer; SCLC: small cell lung cancer; HNSCC: head and neck squamous cell cancer; cHL: classical Hodgkin lymphoma; PMBCL: primary mediastinal large B-cell lymphoma; MSI-H: microsatellite instability-high; dMMR: mismatch repair deficient; RCC: renal-cell carcinoma; TNBC: triple-negative breast cancer.

**Table 2 cancers-12-01475-t002:** Selected biomarkers for novel potential therapeutic strategies to improve PD-1/PD-L1 inibitor efficacy.

Biomarker	Proposed Function	Targeted Agent	Status in Solid Tumors	Reference
Dynamin	Regulation of endocytosis (e.g., EGFR, PD-L1) through Src-FAK signaling	-Dyngo compounds	-Preclinical	[24,25,26,27,28]
		-Prochlorperazine		
		-FAK inhibitors (e.g., defactinib)	-Phase 1-2 studies	
ID1 (Inhibitor of DNA binding I)	-Regulation of cancer stem cells and tumour aggressiveness;	-Gefitinib	-Preclinical	[29,30]
	-Regulation of pathways related to inflammation-associated cell death: activation of necroptosis by triggering activation of RIP1/RIP3/MLKL pathway and minimal effect in inducing pyroptosis;	-ID1 Inhibitors (e.g., AGX-51, pimozide)		
	-ID1 overexpression can be correlated with KRAS and LKB1 mutations.			
β2-adrenergic receptor (β2-AR)	Activation of β2-AR by neurotransmitters, such as norepinephrine, inactivates LKB1, with upregulation of cAMP response element-binding protein (CREB) and interleukin-6 (IL6)	Propranolol	-Phase 1-2 studies	[31]
STAT3	Mediator of tumor-induced immunosuppression	STAT inhibitors (e.g., niclosamide, dihydroartemisinin)	-Phase 1–2 studies	[28,32]
IL-1β	Regulation of tumorigenesis and mediator of immunosuppression through myeloid-derived suppressor cells (MDSCs)	IL-1β inhibitors (e.g., canakinumab, rilonacept, anakinra)	-Phase 1–3 studies	[33,34,35,36,37]
SHP2 (Src homology 2 domain containing phosphatase 2)	Regulation of signaling pathways, in cancer and immune cells, involved in inflammation and tumorigenesis (e.g., RTK, RAS and PD1)	SHP2 inhibitors (e.g., TNO155, RMC-4630, JAB-3068)	-Phase 1–2 studies	[38]
xCT (glutamate-cystine antiporter system)	Glutathione (GSH) synthesis, antioxidant response and ferroptosis	Inhibitors of xCT-GSH pathway (e.g., erastin, iImidazole ketone erastin [IKE], sulfasalazine, dihydroartemisinin [DHA], sorafenib, buthionine sulfoximine [BSO])	-Preclinical (Erastin, IKE);	[28,39,40,41,42,43,44]
			-Phase 1–3 (phase 3 for Sorafenib)	
Acetyl-CoA acetyltransferase (ACAT1)	Cholesterol esterification in T cells	ACAT1 inhibitor (e.g., avasimibe)	-Preclinical	[45]
STING	cGAS-STING pathway: key role in bridging cGAS-STcGAS-STING pathway: key role in cGAS-STING pathway: key role in bridging innate and adaptive anticancer immunity	STING agonists (e.g., ADU-S100, MK-1454, STING agonists (e.g. ADU-S100, MK-1454, GSK3745417)	-Phase 1–2 studies	[46,47,48]
YAP	Master transcriptional regulator involved in multiple cellular functions (activated by NTRK1/NTRK2) and immunosuppression	NTRK or YAP inhibitors (e.g., entrectinib, larotrectinib, repotrectinib, or dihydroartemisinin)	-Phase 1–3 (phase 3 for Entrectinib)	[28,49]
NLRP3	NLRP3 inflammasome: key role in immune response	NLPR3 inhibitors (e.g., OLT1177-dapansutrile, CY-09, tranilast)	-Preclinical	[50,51]
LAG-3	Immune checkpoint receptor modulating T-cell proliferation and activation	LAG-3 inhibitors (e.g., relatlimab)	Phase 1–2 studies	[52,53]

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
