# Peer review of "Non-Small-Cell Lung Cancer Signaling Pathways, Metabolism, and PD-1/PD-L1 Antibodies"

_cancers, 2020, doi:10.3390/cancers12061475_

Round 1

Reviewer 1 Report

This is a comprehensive review about  the description, and analysis of the related pathways, of selected biomarkers that could aid to design novel strategies to improve the PD/PDL! inhibitor efficacy in advanced NSCLC, actually mostly treated with antibodies against PD-1 or PD-L1 alone or in combination with chemotherapy. 

The main aims and points of discussion of the review are very well explained since the abstract.

Infact, the authors state that the review serves to charter diverse treatment solutions, depending on the main altered signaling pathways, in order to have effectual immunotherapy.

The review is very well organized, table 1 introduces the reader to the main biomarker and pathways discussed along the text, and the figures/skecth helps the reader to discern throught the different pathways.

It is also very well presented and represents.

This reviewer would appreciate the mention at line 67 of a commentary (entitled " The Between Now and Then of lung Cancer Chemotherapy and Immunotherapy") by Visconti R et al, IJMS, 2017, doi: 10.3390/ijms18071374.

Reviewer 2 Report

Authors submitted the review article which describes the comprehensive signaling pathways, metabolism and PD-1/PD-L1 antibodies of non-small cell lung cancer (NSCLC) therapy. This review article covers the most of all pathways including the latest reports. There are few such reports that describe comprehensive and new information about the intracellular mechanisms of cancer immune therapy. I appreciate this excellent review article.

Minor suggestion

  1. There are some inconsistency of abbreviation description and spelling. Please carefully revise the manuscript.

Reviewer 3 Report

Immune checkpoint blockade using anti-PD-1 / PD-L1 agents has revolutionized advanced NSCLC therapy. However, a big fraction of patients still do not respond to this therapy or have only shown partial response. Exploration of molecular- and tumor microenvioronmental- drivers, timing, and dosage that could help predict response to anti-PD1 ICB are becoming very critical to determine personalized NSCLC therapy. The authors have performed a commendable job providing a comprehensive study of gene signatures, mutations, altered signaling cascades, and some relatively novel concepts of such as the potential role of neurotransmitters in reprogramming the immune microenvironment in context of anti PD1 ICB and response. However, the study suffers from drawbacks which I have added in the document below. 

Reviewer 4 Report

The present manuscript provides a comprehensive overview on the impact of deregulated signaling pathways in NSCLC on immunotherapy efficacy and the potential novel therapeutic strategies to overcome resistance.

This is a very hot topic. The manuscript is well written and the figures and tables included nicely summarize the concepts described in.

Minor comments:

  • NSCLC harboring known oncogene drivers are usually associated with dismal outcomes when treated with single agent ICIs. However, some recent reports suggest that BRAF-mutant NSCLCs (both V600E and non-V600E) benefit from PD-1/PD-L1 blockage similarly to non-oncogene addicted patients [Dudnik E, et al. J Thor Oncol 2018; Rihawi K, et al. J Thorac Oncol 2019]
  • Discussing the role of concomitant mutations on immunotherapy efficacy, I would also include the findings of some recent studies/reports on ARID1A mutations [Okamura R, et al. J Immunother Cancer 2020; Rizvi NA, et al. WCLC 2019]

Round 2

Reviewer 3 Report

I want to thank the authors for the great rebuttal and review. This work is ready for publication now.